# Mutant alleles differentially shape fitness and other complex traits in cattle

Ruidong Xiang [1,2 ✉], Ed J. Breen[2], Sunduimijid Bolormaa[2], Christy J. Vander Jagt[2], Amanda J. Chamberlain [2], Iona M. Macleod[2] & Michael E. Goddard[1,2]

Mutant alleles (MAs) that have been classically recognised have large effects on phenotype and tend to be deleterious to traits and fitness. Is this the case for mutations with small effects? We infer MAs for 8 million sequence variants in 113k cattle and quantify the effects of MA on 37 complex traits. Heterozygosity for variants at genomic sites conserved across 100 vertebrate species increase fertility, stature, and milk production, positively associating these traits with fitness. MAs decrease stature and fat and protein concentration in milk, but increase gestation length and somatic cell count in milk (the latter indicative of mastitis). However, the frequency of MAs decreasing stature and fat and protein concentration, increasing gestation length and somatic cell count were lower than the frequency of MAs with the opposite effect. These results suggest bias in the mutations direction of effect (e.g. towards reduced protein in milk), but selection operating to reduce the frequency of these MAs. Taken together, our results imply two classes of genomic sites subject to long-term selection: sites conserved across vertebrates show hybrid vigour while sites subject to less long-term selection show a bias in mutation towards undesirable alleles.

[1] Faculty of Veterinary & Agricultural Science, The University of Melbourne, Parkville 3052 VIC, Australia. [2] Agriculture Victoria, AgriBio, Centre for AgriBiosciences, Bundoora, VIC 3083, Australia. ✉email: ruidong.xiang@unimelb.edu.au

Mutant alleles (MAs) that have been classically recognised tend to largely decrease fitness, decrease fitness-related traits and be partially recessive[1–3]. However, the majority of the genetic variance in complex traits is due to mutations with small effects. Do these small-effect mutations show the same characteristics as those classical large-effect mutations? A study in *Escherichia coli* showed that mutations with small effects on fitness tend to be deleterious to protein function[4]. However, how mutations affect complex traits such as body size, health and fertility is unknown.

A better understanding of the consequence of mutations not only updates scientific knowledge but also has practical implications. Domestic cattle support humans by providing food, labour, clothing material and transportation. Today, there are over 4 billion cattle across the world and over ~900 million tonnes of dairy products have been produced annually for human consumption[5]. Genomic selection, which is widely used in animal breeding[6], has been demonstrated to be enhanced by fitting variants with biological priors[7,8]. Therefore, it may be an advantage to also know a priori whether mutations are more likely to increase or decrease traits of interest.

In particular, if a trait is related to fitness, one might expect mutations to be deleterious[2,9]. Therefore, the first objective of this study is to determine whether mutations, defined as the non-ancestral allele (also known as derived alleles) at segregating sites, tend to increase or decrease individual complex traits and whether this depends on the trait's association with fitness.

Traits that are related to fitness typically show inbreeding depression and heterosis caused by directional dominance. That is, fitness decreases with increased inbreeding due to increased homozygosity at loci with recessive deleterious alleles[10]. Conversely, fitness generally increases with heterozygosity[11]. Therefore, directional dominance can be used to link traits to fitness. Here we introduce a test for directional dominance on traits of cattle by estimating the heterozygosity at genomic sites and use this method to identify traits that are associated with fitness. Then, we classify traits showing directional dominance as 'fitness-related traits'.

A likely cause of directional dominance is that mutations tend to be deleterious and partially recessive. However, not all sites in the genome affecting a trait may show this pattern. Our second objective was to test the hypothesis that sites, where the same allele has been conserved across vertebrate evolution, are the most likely to show directional dominance. Therefore, we consider conserved sites and other polymorphic sites in this analysis.

Cattle present a unique opportunity for studying the effects of mutation. The cattle family diverged from other artiodactyls up to 30 million years ago[12]. Modern cattle are derived from at least two different subspecies of wild aurochs, i.e. *Bos primigenius primigenius* (Eurasian aurochs) and *Bos primigenius namadicus* (Indian aurochs), which diverged up to 0.5 million years ago[13–20]. Domestication of *Bos p. primigenius* led to the humpless *Bos taurus* subspecies, which has evolved some highly productive breeds for agriculture, such as the Holstein breed with superior milk productivity. Besides natural selection, dairy cattle breeds experienced very recent and intensive selection for milk production traits[21,22] and stature[23]. Domestication of *Bos p. namadicus* gave rise to the humped *Bos indicus* subspecies which evolved breeds with strong resistance to hot climates, such as Brahman and Gir cattle.

In the present study, we use yak, sheep and camel as outgroup species to assign cattle ancestral alleles for 8 M sequence variants (at 8 M genomic sites). For each of these variants, the alternative to the ancestral allele is the MA. We estimate the effects of the genotypes based on the MA at these 8 M variable sites on 37 traits of 113k cattle from 4 breeds. We also estimate the collective effect of heterozygosity on these traits using either conserved sites or all genomic sites.

If MAs decrease fitness we expect selection to reduce their allele frequency compared with MAs that either have no effect or increase fitness. Therefore, we compared the allele frequency of MAs that increased and decreased each trait. We expanded the analysis of MA frequency to additional breeds of ancient and modern cattle from the 1000 Bull Genomes database[24,25], which provided validation of our results. Additional analyses of MAs with strong effects on milk production traits[26,27] suggested that the direction of phenotypic effects of these MAs correlates with their direction of effects on the expression of genes in milk cells[4,25].

## Results

**Directional dominance at sites conserved across 100 vertebrate species.** To identify traits related to fitness, we have introduced a method to estimate the effect of heterozygosity on 37 traits (described in Supplementary Table 1) recorded in over 100k animals. In total, there were 16,035,443 imputed sequence variants (at 16,035,443 genomic sites) with imputation accuracy $R^2 > 0.4$ and the minor allele frequency (MAF) $> 0.005$ available for variant–trait association analysis. Approximately half of these could be assigned with ancestral alleles and this subset was used for analyses related to MAs (described later). For the analysis of the effect of heterozygosity, we fitted covariates of: (1) the average heterozygosity of sequence variants at 317,279 genomic sites conserved across 100 vertebrate species ($H'_{cons_j}$) and (2) the heterozygosity from variants at the remaining 15,718,164 non-conserved sites ($H'_{non-cons_j}$) simultaneously (see 'Methods'). We observed a significant effect of heterozygosity at conserved sites for the yield of protein (Prot), fat (Fat) and milk (Milk), survival (Surv), fertility performance (Fert), stature (Stat) and angularity (related to slimness and milk yield) (Fig. 1 and Supplementary Fig. 1). For all these traits, heterozygosity at the remaining non-conserved sites ($H'_{non-cons_j}$) was not significant when fitted together with $H'_{cons_j}$. This directional dominance implies that milk production, fertility, survival and stature showed inbreeding depression and heterosis. Therefore, we classified them as fitness-related traits and this directional dominance for these traits was

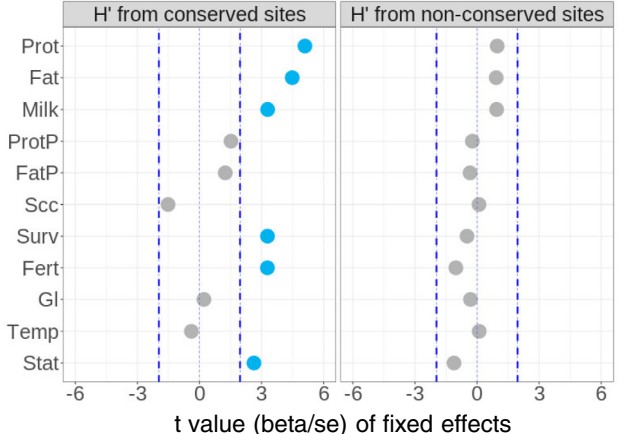

**Fig. 1 Directional dominance at conserved sites ($H'$) for traits of 104k cows.** The beta values and standard errors for each trait were generated using a mixed linear model, fitting a covariate representing $H'$ from 317,279 conserved sites (left panel) and another covariate representing $H'$ from the remaining 15,718,164 non-conserved sites (right panel) together with other fixed effects (e.g. breed). Blue dashed lines indicate $t$ value of −1.96 and 1.96 commonly used to indicate the significance.

predominantly explained by genomic sites conserved across vertebrate species. To be conserved across vertebrate species, mutations at these sites must be deleterious, implying extremely long-term consistent selection for the ancestral allele at these sites.

**Assignment of bovine ancestral and MAs.** To assign the MAs in cattle, we first determined the alternative, ancestral alleles using artiodactyls, including cattle as the focal species (98 global cattle breeds from the 1000 Bull Genomes Project[24,25], Supplementary Table 2) and yak, sheep and camel as outgroup ancestor species (Ensembl 46-mammal sequence data). A probabilistic method[28] was used to assign an ancestral allele for each site mappable between 4 artiodactyl species (see 'Methods'). Out of 42,573,455 equivalent sites between the 4 species, 39,998,084 sites had the ancestral allele assigned with high confidence (probability >0.8). We compared our results with a previous study using different methods[29]. Of the 1,925,328 sites that were assigned ancestral alleles with high confidence in both studies, 1,904,598 (98.7%) sites agreed. However, we assigned ancestral alleles with high confidence to ~10 times more sites than the previous study due to the use of a large sample size and whole-genome sequence data. The full results are publicly available at　https://melbourne.figshare.com/articles/dataset/The_assignment_of_cattle_ancestral_alleles/13546472.

**Biases in trait effects between ancestral and MAs.** We conducted genome-wide association study (GWAS) of 37 traits using over 16 million imputed sequence variants in bulls ($N \sim 9k$) and cows ($N \sim 104k$) separately (see 'Methods'). For 7,910,190 variants where the ancestral allele was assigned, we compared the direction (increase or decrease) of the effect of the MAs on the trait (Supplementary Figs. 2 and 3). The same comparison was also performed for variants at the 202,530 out of 317,279 conserved sites where the ancestral alleles could be assigned. We focus the description of effects on MAs, but a MA that increased the trait is identical to an ancestral allele that decreased the trait.

When analysing all variants and the conserved variants only, for each trait we considered the following three variant categories for systematic comparison: (1) large-effect variants, i.e. GWAS $p$ values ($p_{gwas}$) $< 5e{-}8$ where the effect direction agreed in both sexes; (2) medium-effect variants, i.e. $5e{-}8 <= p_{gwas} < 5e{-}5$ where the effect direction agreed in both sexes; and (3) small-effect variants, i.e. $5e{-}5 <= p_{gwas} < 0.05$ where the effect direction agreed in both sexes. Here the effect size refers to the amount of variance explained by variants which is inversely related to the $p$ value. We used different effect sizes because mutations of small and large effects may be different in their direction of effect. Selecting variants that have the same effect direction between different GWAS populations[30], such as bulls and cows, helped to eliminate variants with spurious trait associations from the comparison. Based on a previous method[30], the True Discovery Rate by Effect Direction (TDRed) of GWAS was calculated between the two sexes across 37 analysed traits for the small-, medium- and large-effect variants resulting in scores of 0.8, 0.98 and 0.99, respectively. On average, each variant from the large, medium and small-effect category explained 0.31% (±0.043%), 0.07% (±0.009%) and 0.015% (±0.0004%) of the variance in cow traits, respectively (Supplementary Table 3).

Based on GWAS results of each trait, we calculated the ratio of the number of variants where the MA increased the trait (positive effect) to the number of variants where the MA decreased the trait (negative effect). Across 37 traits and the three effect-size groups, MAs showed diverse trait effect patterns (Supplementary Fig. 3). Results observed from GWAS were confirmed by BayesR analysis[31], which jointly fitted on average 4.3 million variants per

trait (see 'Methods' and Supplementary Fig. 3). Based on jointly estimated effects for a given set of variants, the significance of the effect direction bias was tested using Kolmogorov–Smirnov to estimate the $p$ value ($p_{ks}$) of the difference in the effect distribution between ancestral and MAs (see 'Methods'). We also tested the significance of bias using linkage disequilibrium (LD)-clumped ($r^2 < 0.3$)[32] variants to calculate the standard error (Supplementary Fig. 4).

In addition, we checked the direction of effects of MAs that had large positive effects and large negative effects on protein yield, fat yield, milk yield, protein% and fat% on the expression of genes within ±1 Mb of these MAs (cis expression quantitative trait locus (eQTL) genes, see 'Methods') in milk cells[26,27]. For four out of five sets of variants where the MA decreased the trait, we found the MA tended to decrease the expression of cis eQTL genes. For another four out of five sets of variants where MAs increased the trait, the MA tended to increase the expression of cis eQTL genes (Supplementary Table 4). These results suggest a correlation between the direction of effects of MAs on milk production traits and the expression of genes in milk cells.

In the following text, we focus on (1) MAs within the large- and small-effect categories for milk production traits as these two sets of MAs showed distinct effect direction patterns (Fig. 2a, b), and (2) MAs associated with other traits, including those with medium or small effects on somatic cell count (Scc, indicative of mastitis, medium effect), survival (Surv, small effect), fertility (Fert, frequency of pregnancy, small effect), gestation length (Gl, medium effect), temperament (Temp, docility, small effect) and stature (Stat, medium effect) (Fig. 2a, b).

The classical model[1–3] predicts that the majority of MAs, or mutations, are deleterious or slightly deleterious. In our study, MAs consistently showed biases towards decreasing protein and fat concentration (Fig. 2a, b and Supplementary Figs. 3 and 4), docility and stature and towards increasing somatic cell count (an indicator of mastitis) and gestation length. Among these traits, only stature showed a significant effect of heterozygosity. For milk yield and protein yield, both of which were classified as fitness-related traits (Fig. 1), the bias in the direction of MA depends on the size of the MA effect. Large-effect MAs tended to decrease milk and protein yield whereas small-effect MAs tended to increase them. A possible explanation is that mutation seldom has a large positive effect on milk protein yield or fertility but small positive effect mutations occur and are increased in frequency by natural or artificial selection.

Also, there was a slight majority of small-effect MAs which tended to increase fertility and survival, both of which were positively related to fitness (Fig. 1). The effects of these sets of MAs is partially due to pleiotropy, i.e. the effect of these MAs on multiple traits (Supplementary Data 1). For instance, while small-effect MAs increasing milk yield decreased fat yield, protein% and fat%, they also increased protein yield. Also, while small-effect MAs increasing fertility increased gestation length, they also increased stature.

The simplest explanation for the bias in the direction of MA effects is that it is due to a bias in the direction of mutation. For instance, that mutation more often leads to a decrease in fat% rather than an increase. However, it is also possible that mutations that decrease fat% are selected and therefore more likely to be discovered than mutations that increase fat%. Below we exclude this possibility by comparing the allele frequency at variants where the MA increased or decreased the trait.

**Allele frequency of MAs in modern and ancient cattle.** Across all variable sites, the allele frequency of MAs was lower than the allele frequency of ancestral alleles (Supplementary Fig. 5). Also,

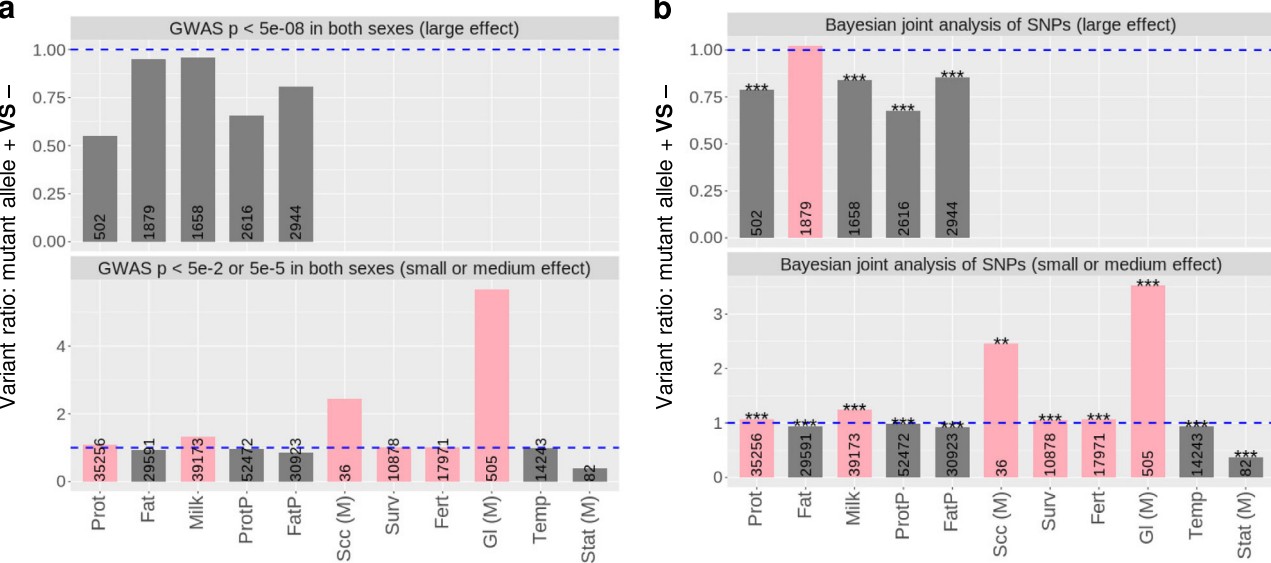

**Fig. 2 The ratio (y-axis) between the number of variants with mutant alleles increasing the trait (+) and the number of variants with mutant alleles decreasing the trait (−).** GWAS effects of mutant alleles are shown for all variants (**a**). BayesR joint effects of mutant alleles from the same variants in **a** are shown for all variants (**b**). Pink colour: the majority of variants with mutant alleles tend to increase the trait (taller than the blue-dashed line). Dark grey: the majority of variants with mutant alleles tend to decrease the trait (shorter than the blue-dashed line). Numbers in bars: total number of variants significant at the given threshold. Stars: $p$ value for the significance of the difference in the distribution of BayesR effects between ancestral and mutant alleles, *$p < 0.05$, **$p < 0.01$, ***$p < 0.001$. For somatic cell count (Scc), gestation length (Gl) and stature (Stat), the results are from medium-effect (M) variants and the full results are shown in Supplementary Fig. 3.

the frequency of MAs at conserved sites (0.27) was lower than the frequency of MAs across all sites (0.32). This is consistent with selection for the ancestral allele which is necessary to maintain conservation of the same allele across vertebrates.

We grouped variants based on their MA reducing (MA−) or increasing the trait (MA+) and compared their allele frequency in over 110k Holstein, Jersey, crossbreds and Australian Red bulls and cows (Fig. 3a, b). To account for LD, we estimated the error of MA frequency based on LD-clumped ($r^2 < 0.3$)[32] variants. As an external validation, we also considered this analysis in a selection of 7 subspecies/breeds of 1720 ancient and modern cattle from the 1000 Bull Genomes Project[24,25] (Fig. 3c, d).

For fat%, protein%, docility and stature MAs that increased the trait had higher allele frequency than MAs that decreased the trait. For somatic cell count and gestation length, the reverse was true. That is, MAs increasing somatic cell count and gestation length had lower allele frequency than MAs that decreased the trait (Fig. 3b, d). Thus, although MAs more commonly decreased fat% than increased it, the allele frequency was higher at sites where the MAs increased fat%. This implies that selection has acted against MAs that decreased fat% or favoured MAs that increased fat%. Consequently, the higher incidence of MAs that decreased fat% cannot be due to selection favouring them but must be due to the mutation more often resulting in an allele that decreased fat% than increased it. Comparing results in Figs. 2 and 3 shows that this is the usual pattern—the more common direction of effects of mutation generated alleles that were selected against and hence had a reduced allele frequency.

For other traits, the results were less clear-cut. For milk yield, the majority of MAs of large effect tended to decrease the trait (Fig. 2b). Interestingly, these large-effect milk-decreasing MAs, which were deleterious, had a higher frequency than those MAs increasing milk yield (Fig. 3a, c). On the other hand, the majority of MAs of small effect tended to increase milk yield (Fig. 2b). Yet, these small-effect MAs that increased milk yield were at a lower frequency than MAs that decreased milk yield (Fig. 3a, c). Note

that milk yield is negatively correlated with fat% and protein% ($r = −0.83$ and $−0.78$, respectively).

**Selection of trait-associated MAs in modern and ancient cattle.** The above results for MA frequency at trait-associated variants imply selection. The selection could have been consistent across breeds which would limit the divergence of allele frequency between breeds or it could have been different between breeds leading to divergence in allele frequency. We compared the average of Wright's fixation index ($\overline{F_{ST}}$), for MA+ variants and MA− variants calculated using dairy cattle (Fig. 4a, b) and ancient and modern cattle (Fig. 4c, d). To account for LD, we estimated the error of $\overline{F_{ST}}$ based on LD-clumped ($r^2 < 0.3$)[32] variants.

In general, variants associated with milk production traits (including somatic cell count, Fig. 4a) showed higher than average $F_{ST}$ among dairy breeds implying divergent selection, while variants associated with other traits, including survival and fertility, tended to have below-average $F_{ST}$ indicating convergent selection (Fig. 4b). $\overline{F_{ST}}$ for gestation length was below average especially for MA+, probably due to selection against mutations that increased gestation length in all breeds (Fig. 4d).

Among ancient and modern cattle, $\overline{F_{ST}}$ was high for both MA + and MA− variants for stature indicating divergent selection for height (Fig. 4b). The allele frequency of MAs that decreased height was the least frequent in Holstein cattle and was most frequent in Tibetan cattle living at high altitudes and Angus cattle selected for beef production (Fig. 3d). This suggests that the direction of selection could vary across cattle breeds under different environmental conditions and/or artificial selection.

## Discussion
For some traits (e.g. survival, fertility), we expect that an increase in the trait leads to an increase in fitness. It is these traits that typically show heterosis and inbreeding depression due to

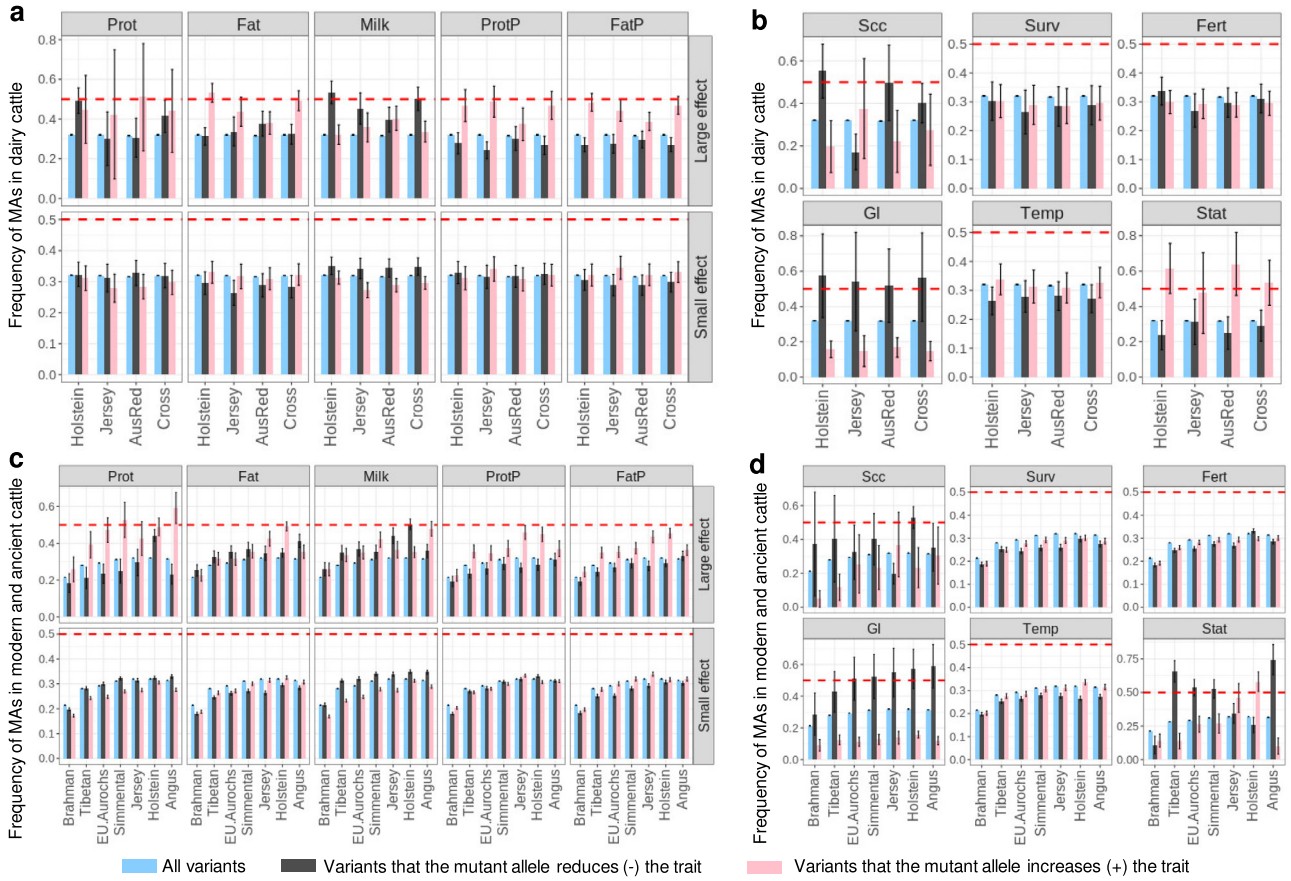

**Fig. 3 The allele frequency of mutant alleles (MAs) in cattle.** The average frequency of variants associated with different traits is shown with standard error bars based on LD-clumped variants. All variants included the 7.9 M variants where mutant alleles were assigned. The red dashed line represents the frequency of 0.5. In the dairy cattle section (**a**, **b**), 90,627 Holstein, 13,465 Jersey, 3358 Australian Red (AusRed) and 4649 crossbreds were used. In the ancient and modern cattle (**c**, **d**), 210 Brahman, 25 Tibetan, 10 Eurasian Aurochs, 242 Simmental, 95 Jersey, 840 Holstein and 287 Angus were used. For **b**, **d**, results for survival (Surv), fertility (Fert) and temperament (Temp) were from small-effect MAs while results for somatic cell count (Scc), gestation length (Gl) and stature (Stat) were from medium-effect MAs.

directional dominance. The simplest explanation for these observations is that mutations at sites affecting the trait tend to reduce the trait and be partially recessive. However, our results show that it was not all sites affecting these traits that showed directional dominance but only those where the same allele was highly conserved across vertebrates. This result explains why the mutations tend to lead to a decrease in the trait—long-term selection has nearly fixed the favourable allele and so any mutation will cause a decrease in the trait and in fitness. We partially confirm this explanation by finding that mutations for these traits (milk and protein yield, stature but not fertility and survival) do tend to decrease the trait although, for milk and protein yield, it was only mutations of large effect for which the effects tended to be negative. This long term selection cannot be directly on traits involving lactation since the same allele was conserved in vertebrates other than mammals.

For other traits, we expected that an intermediate value leads to the highest fitness. For instance, too high or too low a fat% in milk might be detrimental to the fitness of the mother or the infant or both. These traits do not typically show inbreeding depression or heterosis. The fittest allele might vary between species and environments. Therefore, one might expect that mutations are equally likely to increase or decrease the trait. However, that is not what we found: for fat% and protein% mutations tended to decrease the trait whereas for SCC and gestation length they tended to increase the trait. We hypothesise

that at some of the genomic sites affecting these traits selection has been consistent enough in mammals, or at least in cattle so that mutations cause a decrease in fat% and protein% and an increase in mastitis or SCC and gestation length (leading to difficulty calving). This hypothesis was supported by our finding that selection decreased the allele frequency of these mutations. This low allele frequency was not only seen in dairy cattle but in beef breeds and *B. indicus* breeds.

The findings on individual traits can be unexpected due to pleiotropy. That is, mutations affect multiple traits. There are mutations at *DGAT1* and *GHR* loci that increase milk yield but decrease fat% and protein% (Supplementary Fig. 6). These are only at appreciable frequency in domesticated cattle, especially breeds artificially selected for milk volume. Their low allele frequency in other breeds and species suggest that natural selection acts against the mutation thus increasing fat% and protein% but decreasing milk yield. Similarly, there is a negative genetic correlation between milk yield and fertility so mutations that increase milk yield might be favoured despite their negative effect on fertility. MAs decreasing fertility tended to be most frequent in the Holstein breed (Fig. 3d), perhaps because these alleles tended to increase milk yield and stature.

For milk, fat and protein yield, the results differed between mutations of large and small effects. Mutations with a large effect on milk protein yield more often decreased protein yield than increased it perhaps because the physiology supporting milk

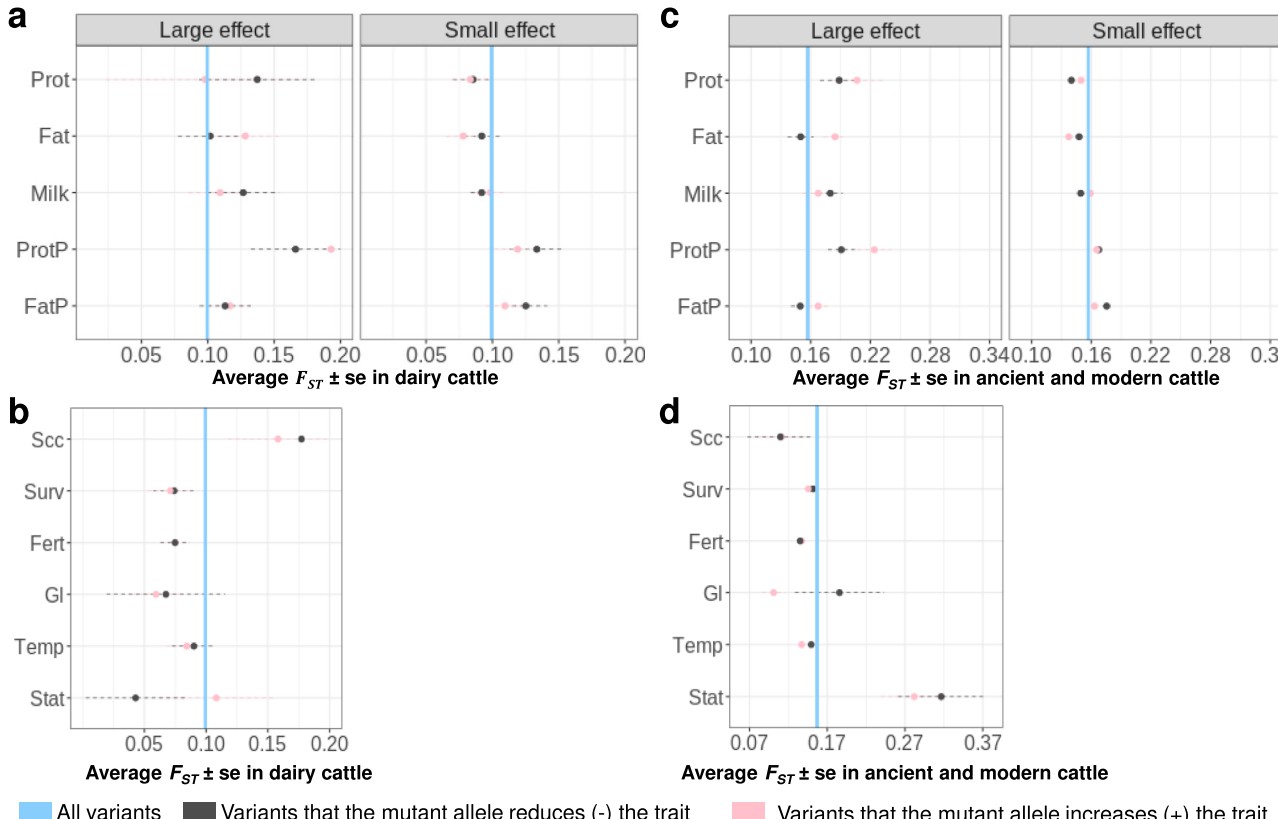

**Fig. 4 Selection (average Wright's fixation index $\overline{F_{ST}}$) of variants with mutant allele that increases or decreases the trait in dairy cattle and ancient and modern cattle.** The $\overline{F_{ST}}$ is shown as dots with its standard error bars estimated using LD-clumped variants. The blue line represents the $\overline{F_{ST}}$ for 7.9 M variants analysed (0.1 ± 4.3e−05) in dairy cattle in **a**, **b**; and $\overline{F_{ST}}$ = 0.157 ± 5e−05 in ancient and modern cattle in **c**, **d**. In the dairy cattle section (**a**, **b**), 90,627 Holstein, 13,465 Jersey, 3358 Australian Red (AusRed) and 4649 crossbreds were used. In the ancient and modern cattle (**c**, **d**), 210 Brahman, 25 Tibetan, 10 Eurasian Aurochs, 242 Simmental, 95 Jersey, 840 Holstein and 287 Angus were used. For **b**, **d**, results for survival (Surv), fertility (Fert) and temperament (Temp) were from small-effect MAs, while results for somatic cell count (Scc), gestation length (Gl) and stature (Stat) were from medium-effect variants.

protein synthesis has been optimised in part at least. Mutations with a small effect on protein yield were almost equally likely to increase or decrease yield perhaps because natural selection favours an intermediate level of milk protein yield because too high a yield drains the cow of nutrients needed for maintenance and reproduction.

Effects of MAs on phenotypes might be mediated by their effects on gene expression. Based on *cis* eQTL data[26], we found that MAs with large effects on milk production traits had direction of effects that were correlated with their direction of effects on gene expression in milk cells. This result shows that the effect direction of MAs on gene expression may also have systematic biases and this may be related to their effects on phenotypic traits. Future studies with larger sample size and more tissues for eQTL mapping may update our understanding of the MA effects on molecular phenotypes.

The selection which we have observed affecting the frequency of mutations of positive and negative effect could be both natural selection acting over a long period before and since the domestication of cattle, and artificial selection acting over the past 10,000 years and, more intensely, over the past ~100 years in dairy cattle. Artificial selection may differ between breeds and generate high $F_{ST}$ between breeds. For fat%, protein% and stature at least one class of mutation was more common than random mutations and the overall $F_{ST}$ between breeds tended to be high. Our analysis also highlighted some specific breeds. For example, the selection of variants associated with somatic cell count led to

high $F_{ST}$ among dairy cattle but low $F_{ST}$ in our other breeds. Holstein cattle have been selected to be tall[23] and this is reflected in the low frequency of MAs decreasing stature in Holstein. On the other hand, the high frequency of MAs decreasing stature in Tibetan cattle (Fig. 4d) may be due to its adaptation to high altitude[33].

Although mutation was biased in its effect on some traits, the bias was small for most traits. That is, mutations decreasing protein yield were only slightly more common than mutations that increased protein yield. Also, although conserved sites explain directional dominance and are enriched for polymorphisms affecting complex traits[34], they do not explain the majority of the genetic variance. That is, there are many sites affecting traits, such as milk yield and stature, at which the allele carried varies between species implying that the fittest allele varies depending on the environment and the background genotype of the species.

The sequence variants associated with a complex trait are not necessarily causal but likely to be in high LD with the causal variants. This tends to dilute the signal that might be discovered if causal variants were used. However, variants in high LD may share a similar evolutionary history and therefore show some of the same characteristics. We used BayesR which jointly fitted variants and LD-clumping to account for LD. However, we acknowledge that we cannot completely remove the effects of LD on our results. Therefore, future studies with even larger sample sizes, e.g. ~1 million, may update our results.

Genomic selection[35], used in the breeding of livestock and crops, estimates the genetic value of individuals for traits of interest from the alleles they carry at genetic markers, such as single-nucleotide polymorphisms (SNPs). The equation predicting genetic value uses the effect of each SNP on the trait estimated in a training population. The best methods treat the SNP effects as random variables drawn from a prior distribution. To date it has been assumed that the effects of a mutation are equally likely to be positive or negative on the trait but, if it was known that one direction of effect was more likely, this could be built into the prior distribution resulting in an increase in the accuracy with which genetic value is predicted.

In conclusion, our results support a hypothesis which provides a new picture of the effects of mutation and selection on mammalian complex traits. Directional dominance, which causes heterosis and inbreeding depression, is characteristic of loci where mutations decrease the trait and fitness and this pattern has been consistent over the evolution of vertebrates. More recent selection, although not causing directional dominance, leads to a bias in the direction of mutation because the mutation results in an allele which is less fit than the ancestral allele and tends to affect a complex trait in a consistent direction. This hypothesis, if supported by future research, adds to our understanding of the evolution of complex traits and has practical value in the artificial selection of livestock and other species.

## Methods

**Data preparation for calling bovine ancestral alleles**. The assignment of bovine ancestral alleles was based on a model comparison of alleles from cattle with alleles from outgroups of yak (*Bos grunniens*), sheep (*Ovis aries*) and camel (*Camelus dromedarius*). According to the evolutionary relationships reported previously[12], among ruminants, yak is an outgroup species closely related to cattle, while sheep is less closely related to cattle than yak. Goat is equivalent to sheep in its relationship to cattle, but we chose sheep in the current study. Camel without a rumen is distantly related to ruminant cattle, as they are artiodactyls. For the cattle species, we used whole-genome sequence data of 98 breeds, one individual per breed, from Run 7 of the 1000 Bull Genomes Project[24,25]. Only those whole-genome sequence samples with coverage >10× were selected and if multiple individuals were found for a breed, the whole-genome sequence sample with the highest coverage was chosen. Both *B. taurus* and *B. indicus* subspecies were included (Supplementary Table 2). The pre-processing of sequence reads and alignment of sequence data was done by project partners using the standard 1000 Bull Genomes Project pipeline[25,36]. Only BAM files from 1000 Bull Genomes partners are collected and processed by the consortium. The latest published data from the 1000 Bull Genomes Project (1832 samples) can be found at https://www.ebi.ac.uk/eva/?eva-study=PRJEB42783. The details of variant calling can be found in ref. [37]. Briefly, Genome Analysis Toolkit (GATK v.3.8)[38] was used for variant calling. Variants from the GATK VQSR (Variant Quality Score Recalibration) 99.90 to 100.00 Tranche for SNP and INDEL were excluded, and Beagle v.4.0[39] was used to impute variants with sporadic missing genotypes. Whole-genome sequence data in VCF format for these 98 cattle, as a subset from the 1000 Bull Genomes Project database, was generated for further analysis.

For the outgroup species (to determine ancestral alleles), we used whole-genome sequence data of 46 mammals stored in the Multiple Alignment File generated by Ensembl EPO pipeline[40]. The 46-mammal EPO Multiple Alignment File was downloaded. Then the software WGAbed[41] was used to retrieve sequence data for cattle, yak, sheep and camel in bed file format. Only sites with sequence data available in at least one outgroup species were kept. Using the cattle coordinates in the 4-species WGAbed files, the sequence data of the outgroup species were matched with the 98 cattle. As a result, 42,573,455 sites found in the 98 cattle and in at least one outgroup species were found. Sequence data on these 42,573,455 sites across 4 species were used to determine the bovine ancestral alleles.

**Probabilistic determination of bovine ancestral alleles**. We used the method proposed by Keightley et al.[28] with the model choice of the Kimura two-parameter (K2), which accounts for allele frequency of the focal species to determine the probability of an allele being ancestral at each available site. The method was implemented in estsfs[28] and the K2 model was chosen due to its equivalent accuracy to other models but better computation efficiency. As described above, the sequence data of three outgroup species were used. The order of phylogenetic tree topology was cattle → yak → sheep → camel. As required by the software, allele counts of A, C, G and T were determined for the focal species (cattle) and for out species at each available site. For cattle, the total allele count for each site was 196

(98 × 2). For each outgroup species, the total allele count for genome sequences at each site was up to 1. Missing sequence data in the outgroup species were treated as 0 counts. For each site, estsfs produced a probability ($P_{ancs}$) of the major allele in the focal species being ancestral. We then determined alleles which were major at a site with $P_{ancs} > 0.8$ or those alleles that were minor at a site with $P_{ancs} < 0.2$ to be ancestral. For those sites where the major or minor alleles could not be determined but the $P_{ancs} > 0.8$ or <0.2, the cattle allele with the highest frequency in the 3 out species was assigned ancestral. The rest of the sites were determined as ambiguous as no clear ancestral alleles could be determined. The detailed results of ancestral alleles for those 42,573,455 sites across 4 species and the probability of the alleles being ancestral or ambiguous is publicly available at: https://melbourne.figshare.com/articles/dataset/The_assignment_of_cattle_ancestral_alleles/13546472.

**Sequence variants under conserved sites across 100 vertebrate species**. The variant selection followed previous procedures[34]. Briefly, conservation was determined by the criteria of PhastCon score[42] >0.9 based on the sequence data of those 100 species. The choice of 0.9 as the cutoff was arbitrary. However, since PhastCons score ranges from 0 to 1, this cutoff kept relatively highly conserved sites. Also, in a previous study[34], cattle variants from sites with PhastCon score >0.9 were highly enriched for the heritability of cattle traits (occupying 2% of the genome to explain up to 42% heritability of traits). The conserved sites were primarily determined using the human genome coordinates (hg38) and were lifted over to the bovine genome ARS-UCD1.2 using the LiftOver software[43] with a lift-over rate >92%. In total, 317,279 variants in the current study were assigned as the conserved variants.

**Animals and phenotypes for variant–trait association analysis**. Data were collected by farmers and processed by DataGene Australia (http://www.datagene.com.au/) for the official May 2020 release of National breeding values. No live animal experimentation was required. DataGene provided the bull and cow phenotypes as de-regressed breeding values or trait deviations for cows and daughter trait deviations for bulls (i.e. progeny test data for bulls). DataGene corrected the phenotypes for herd, year, season and lactation following the procedures used for routine genetic evaluations in Australian dairy cattle. Phenotype data included a total of 8,949 bulls and 103,350 cows from DataGene, including Holstein (6886♂/87,003♀), Jersey (1562♂/13,353♀), cross-breed (36♂/5037♀) and Australian Red dairy (265♂/3379♀) breeds. In total, 37 traits were studied that related to milk production, mastitis, fertility, temperament and body conformation (Supplementary Table 1). Larger trait values of fertility (Fert), ease of birth (Ease), temperament (Temp), milking speed (MSpeed) and likeability (Like) meant poor performances in the farmer evaluation of these traits. Therefore, to assist the interpretability of the study, we have reversed the trait scale so that higher values of Fert, Ease, Temp, MSpeed and Like indicate increased fertility performance (calving frequency), labour ease, docility, milking speed and the overall preference as a dairy cow (Supplementary Table 1). This correction only affected the reported effect direction of the MA.

**Genotype data for association analysis**. The genotypes used in the current study included a total of 16,035,443 bi-allelic sequence variants with Minimac3[44,45] imputation accuracy $R^2 > 0.4$ and the MAF > 0.005 in both sexes. Most bulls were genotyped with a medium-density SNP array (50 K: BovineSNP50 Beadchip, Illumina Inc.) or a high-density SNP array (HD: BovineHD BeadChip, Illumina Inc.) and most cows were genotyped with a low-density panel of approximately 6.9k SNPs overlapping with the standard-50K panel. The low-density genotypes were first imputed to the Standard-50K panel and then all 50 K genotypes were imputed to the HD panel using Fimpute v3[34,46]. Prior to sequence imputation, the HD genotypes were converted to forward sequence format. Then all HD genotypes were imputed to sequence using Minimac3 with Eagle (v2) to pre-phase genotypes[45,47]. The reference set for imputation included sequences of 3090 *B. taurus* animals from Run7 of the 1000 Bull Genomes Project[24] aligned to the ARS-UCD1.2 reference bovine genome[25,48]. The accuracy of the sequence data for individual animals in the 1000 Bull Genomes Project is routinely checked against their own high-density SNP array genotypes and the concordance has been >95%[37]. The empirical accuracy of imputation to sequence using the 1000 Bull Genomes project has been routinely tested for dairy breeds: for example, in Holsteins, the average correlation between imputed and real sequence variants was 0.92 to 0.95 using Run5 of the 1000 Bull Genomes project ($N = 1577$)[49]. Therefore, we believe our imputed data are more accurate: first, because the number of reference animals has almost doubled and second because in our study we impose a Minimac3 $R^2$ filter to remove poorly imputed variants. A Minimac3 $R^2$ threshold of 0.4 was used because our in-house tests demonstrate that this is approximately equivalent to an empirical imputation accuracy (correlation) of 0.85.

**Genome-wide association studies**. The above-mentioned traits were analysed one trait at a time in each sex with linear mixed models using GCTA[50]:

$$\mathbf{y} = \mathbf{mean} + \mathbf{breed} + \mathbf{bx} + \mathbf{a} + \mathbf{error} \qquad (1)$$

where **y** = vector of phenotypes for bulls or cows, **breed** = three breeds for bulls, Holstein, Jersey and Australian Red and four breeds for cows (Holstein, Jersey,

Australian Red and MIX); **bx** = regression coefficient $b$ on variant genotypes **x**; **a** = random polygenic effects $\sim N(0, \mathbf{G}\sigma_g^2)$ where **G** = genomic relatedness matrix based on all variants and $\sigma_g^2$ = random polygenic variance; **error** = the vector of random residual effects $\sim N(0, \mathbf{I}\sigma_e^2)$, where **I** = the identity matrix and $\sigma_e^2$ the residual variance. The purpose of fitting breeds as fixed effects together with the GRM in the model was to have strong control of the population structure which may cause spurious associations between variants and phenotype. The construction of GRM followed the default setting (--make-grm) in GCTA[50]. In the results of GWAS (the same for results from BayesR described below), we matched the allele with which the beta was estimated to the MA. Then this effect is defined as the effects of the MA. Note that, for a variant, the effect of a MA is identical to $-1\times$ the effect of the ancestral allele.

**Bayesian mixture model analysis**. In the above-described GWAS, sequence variants, many of which are in high LD, were analysed one at a time. In order to assess variant effects and account for LD, we fitted selected variants jointly in BayesR[31]. For each trait, variants that showed the same sign between bulls and cows (regardless of $p$ value) and could be assigned with an ancestral allele were analysed with BayesR. Across 37 traits, the number of variants analysed ranged from 3,961,180 to 4,737,492. To reduce the computational burden of BayesR, we estimated the joint effects of these variants for each trait in bulls. BayesR models the variant effects as mixture distribution of four normal distributions including a null distribution, $N(0, 0.0\sigma_g^2)$, and three others: $N(0, 0.0001\sigma_g^2)$, $N(0, 0.001\sigma_g^2)$, $N(0, 0.01\sigma_g^2)$, where $\sigma_g^2$ was the additive genetic variance for the trait. The starting value of $\sigma_g^2$ for each trait was estimated using GREML implemented in MTG2[51] with a single genomic relationship matrix made of all 16 M sequence variants. The statistical model used in the single-trait BayesR was:

$$\mathbf{y} = \mathbf{Wv} + \mathbf{Xb} + \mathbf{e} \tag{2}$$

where **y** was a vector of phenotypic records; **W** was the design matrix of marker genotypes; centred and standardised to have a unit variance; **v** was the vector of variant effects, distributed as a mixture of the four distributions as described above; **X** was the design matrix allocating phenotypes to fixed effects; **b** was the vector of fixed effects of breeds; **e** = vector of residual errors. As a result, the effect $v$ for each variant jointly estimated with other variants were obtained for further analysis.

**The difference in effect distribution between ancestral and MAs**. For an analysed variant, one allele is ancestral and then the other is mutant. If there is a bias in effect direction in ancestral alleles or MAs in a given set of variants, the effect distribution of the ancestral and MAs would be different. For example, for a given set of variants, if their MAs had a bias in effect direction towards increasing the trait, their ancestral alleles would have a bias in effect direction towards decreasing the trait. This then would create a difference in effect distribution between mutant and ancestral alleles. We tested if the distribution of the effect of ancestral alleles estimated from BayesR was significantly different from that of MAs using the two-sample Kolmogorov–Smirnov test implemented by ks.test() in R v3.6.1. The coding was ks.test(a,m) where a was the vector of variant effects based on the ancestral alleles and m was a vector of variant effects based on the MAs. To be more conservative, we also tested the significance of biases using LD-clumped ($r^2 < 0.3$ within 1 Mb windows) variants with small, medium and large effects using default settings in plink1.9[32].

**Heterozygosity of individuals at conserved sites**. It is widely accepted that higher genomic heterozygosity is linked to gene diversity, therefore, fitness. However, it is not clear at which set of genes or variants heterozygosity is more related to fitness. Also, the simple estimation of heterozygosity, i.e. assigning allele counts of 0 or 2 as homozygous and 1 as heterozygous, leads to biases as the estimation is not independent of additive effects (illustrated later). Our previous work showed that conserved sites across 100 vertebrate species significantly contribute to trait variation[7,34] and it is also logical to assume that mutations at conserved sites tend to have strong effects on fitness. Therefore, we first partitioned the genome into 317,279 conserved and 15,718,164 non-conserved variants. Then we re-parameterised the genotype allele count for each variant commonly used to model the dominance deviation, so that the estimation of dominance deviation is independent of the additive effects. We focussed on cows because their traits were largely measured on themselves, contrasting to bull traits that were based on their daughters' traits. We estimated the variant-wise sum of the re-parameterised allele count value for dominance deviation, which was later termed as $z'_{D_i}$ for each variant $i$ in cows. The sum was averaged by the number of variants and this average value based on re-parameterised dominance allele count for the individual $j$ was termed as $H'_j$ to represent the individual heterozygosity. We estimated the individual heterozygosity from conserved sites ($H'_{\text{cons}_j}$) and non-conserved sites ($H'_{\text{non-cons}_j}$) and these computations are specified in the following text.

According to quantitative genetics theory[52–54], the genetic value ($G'$) of an individual can be partitioned into the mean ($\mu$), additive genetic value ($A$) arising from additive effect ($a$) and dominance genetic value ($D$) arising from dominance deviation ($d$). At a single locus, let the allele frequency of the three genotype classes

of AA, AB and BB be $p^2$, $2pq$ and $q^2$, respectively. In a simple genetic model, the genetic value can be decomposed as:

$$G' = \mu + A + D + e = \mu + x_{A_i}a + z_{D_i}d + e \tag{3}$$

where $x_{A_i}$ was the allele count for genotype AA, AB and BB for locus or variant $i$, which were usually coded as 0, 1 and 2, respectively, to represent the additive component, and $z_{D_i}$ was usually coded as 0, 1 and 0, for genotype AA, AB and BB for variant $i$, respectively, which differentiates the homozygous and heterozygous to represent the dominance component. Therefore, in the simplest form, the genome-wide heterozygosity of the individual $j$ can be calculated as:

$$H_j = \sum_i^N z_{D_i}/N \tag{4}$$

where $H_j$ is the simple genome-wide heterozygosity of individual $j$ and $N$ is the total number of variants. Note that such calculation of $H_j$ can also be used to derive inbreeding coefficient, where $I_j = (\sum_i^N 2p_iq_i) \times H_j$. $I_j$ was the inbreeding coefficient for the $j$th individual.

In Eq. 3, however, due to the non-zero correlation between $x_A$ and $z_D$ under Hardy–Weinberg equilibrium (HWE), the estimation of $a$ and $d$ is not independent, i.e. $\text{cov}(x_A, z_D) = 2p(1 - p)(1 - 2p) \neq 0$ under HWE. This then resulted in the estimation of $H_j$ not being independent of the additive components. Therefore, we proposed to re-parameterise this model to estimate $a$ and $d$ independently.

According to Falconer[53] at this locus, the additive effects can be derived using the regression of genetic value on the number of A alleles, where $A'_{AA} = 2q \times \alpha$, $A'_{AB} = (p - q) \times \alpha$ and $A'_{BB} = -2p \times \alpha$. $A'$ is the re-parameterised additive genetic value and $\alpha$ is the allele substitution effect: $\alpha = a + (p - q)d$. Because the dominance deviation is the difference between the genetic value and the mean plus the additive value, the dominance effects can be derived as $D'_{AA} = -2p^2 \times d$, $DD'_{AB} = 2pq \times d$ and $D'_{BB} = -2q^2 \times d$. $D'$ is the re-parameterised dominance genetic value. Therefore, Eq. 3 can be re-parameterised as:

$$G' = \mu + A' + D' + e = \mu + x'_{A_i}\alpha + z'_{D_i}d + e \tag{5}$$

where $x'_A$ was coded as $2q$, $p - q$ and $-2p$ for genotype of AA, AB and BB of variant $i$, respectively, to represent the additive component and $z'_D$ was coded as $-2p^2$, $2pq$, $-2q^2$ for genotype of AA, AB and BB of variant $i$, respectively, to represent the dominance component. Such re-parametrisation has the following features: (1) The covariance between the additive and dominance effects is zero; (2) the variance of the additive effects gives the additive variance; and (3) The variance of the dominance deviations gives the dominance variance. Equation 5 then leads to:

$$H'_j = \sum_i^N z'_{D_i}/N \tag{6}$$

where $H'_j$ was the re-parameterised genome-wide heterozygosity for individual $j$, $z'_D$ was $-2p^2$, $2pq$ and $-2q^2$ for the genotype of AA, AB and BB of variant $i$ and $N$ was the total number of variants. We then applied Eq. 6 to conserved and non-conserved variants to estimate individual heterozygosity from conserved sites ($H'_{\text{cons}_j}$) and non-conserved sites ($H'_{\text{non-cons}_j}$). We then fitted $H'_{\text{cons}_j}$ and $H'_{\text{non-cons}_j}$ as fixed effects together with the fixed effects of breed jointly in GREML similar to Eq. 1. The difference was that there is no fixed effect of variants but more fixed effects due to the fitting of $H'_{\text{cons}_j}$ and $H'_{\text{non-cons}_j}$. The GREML analysis used the implementation with MTG2[51].

**MA frequency and $F_{ST}$ in different breeds/subspecies**. Two sets of data were used for this analysis. The first data set was the Australian dairy cattle (8949 bulls and 103,350 cows, Holstein, Jersey, Australian Red and crossbreds) used for GWAS as described above. The second data set used for the analysis of MA frequency and $F_{ST}$ was the curated whole-genome sequence data of 1720 cattle from the 1000 Bull Genomes database (Run 7)[24,25], which we refer to as modern and ancient cattle. Samples that met the quality criteria of the 1000 Bull Genomes project were selected and they included 210 Brahman, 25 Tibetan, 10 Eurasian Aurochs, 242 Simmental, 95 Jersey, 843 Holstein and 295 Angus. Genome sequences from 6 Gir and 12 Nellore cattle from the 1000 Bull Genomes database were also analysed to support the results of MA frequency of *B. indicus*. Additional information on these 1720 animals including related accession numbers (if available) can be found in Supplementary Data 2. The ancient genome data were part of the project of Verdugo et al.[18] who processed and published the original data (PRJEB31621 at European Nucleotide Archive). These data were collected by Run 7 of the 1000 Bull Genomes Project and processed by its standard pipeline[36].

Sequence data at 7,910,190 variants assigned with MAs were retrieved for these animals to make a plink (v1.9) binary genotype file. The A1 allele of the plink genotypes was set to the MA and its frequency was calculated using the '--freq' function for different selections of populations and variant sets. Average MA frequency and the standard error were calculated for different selections of variants, e.g. variants with MAs increasing or decreasing traits. Standard errors for frequency and $F_{ST}$ (described below) were all estimated using LD-clumped variants

in the same procedure in plink[32] as described above. For variants associated with milk production traits, i.e. the yield of milk protein, fat and milk and percentage of protein and fat, we selected variants with large (GWAS $p$ value $< 5e{-}8$ in both sexes) and small (GWAS $p$ value $< 5e{-}2$ and $p$ value $> 5e{-}5$ in both sexes) effects to focus on. For other trait-associated variants, the group with the largest effects available were selected for this comparison. For example, for stature, there were no variants with $p$ value $< 5e{-}8$ in both sexes, we then selected the medium-effect variants (GWAS $p$ value $< 5e{-}5$ and $p$ value $> 5e{-}8$ in both sexes). For fertility, there was no variants with $p$ value $< 5e{-}5$ in both sexes, we then selected the small-effect variants (GWAS $p$ value $< 5e{-}2$ and $p$ value $> 5e{-}5$ in both sexes) for the comparison. Average MA frequency and the standard error were also calculated for all 7.9 M variants analysed as the baseline. The analysis procedure for allele frequency on the Australian dairy cattle was applied to these 1000 Bull Genomes individuals.

To quantify the trait variance explained by variants with large-, medium- and small-effect from GWAS results, the following formula was used:

$$Vp_i = \chi_i^2/N = t_i^2/N = \left(\frac{b_i}{\text{se}}\right)^2/N \tag{7}$$

where $Vp_i$ was the proportion of phenotypic variance explained by variant$_i$, $\chi_i^2$ was the chi-square value of the effect of variant$_i$ that equals to the square of $t$ value ($b_i$/se), $t_i^2$, of the effect of variant$_i$ from GWAS; $N$ was the sample size of the GWAS; $b_i$ was the GWAS beta of variant$_i$ and se is the standard error of $b_i$. Equation 7 equals the formula $V_p = 2p_i(1-p_i)*b_i^2$, where $p_i$ was the MAF of variant$_i$. However, Eq. 7 is not affected by the MAF of the variant. Then the average $Vp_i$, or $\overline{Vp_i}$, for variants with large-, medium- and small-effect, was estimated for each trait.

With the same plink binary genotype file described above and the population structure for dairy cattle (four dairy breeds) and for ancient and modern cattle (seven breeds/subspecies), GCTA[50] was used to calculate the $F_{ST}$ value with the method described in Weir[55] with the option of '--fst' and '--sub-pop'. The average $F_{ST}$ value with standard errors was then calculated for different selections of variants in the same fashion for selecting variant groups to compare the MA frequency as described above.

**_cis_ eQTL in milk cells**. This analysis was based on 105 Holstein cattle that had RNA-seq data in milk cells described and published previously (NCBI SRA SRP111067)[26,27]. The raw reads of these data were aligned to the ARS-UCD1.2 reference bovine genome using STAR[56]. Qualimap 2[57] and RseQC[58] were used to check data quality. FeatureCount[59] was used to extract gene counts and the voom[60] normalised counts were used in the following analyses. The normalised gene expression was analysed as phenotypes in the same GWAS model as Eq. 1 using GCTA, except that there were no breed effects (all animals are Holstein) but were other fixed effects of Experiment, Days in Milk, first PC and second PC extracted from the expression count matrix. Variants analysed were those that had large positive effects and large negative effects ($p_{\text{gwas}} < 5e{-}8$) on protein yield, fat yield, milk yield, protein% and fat%. For these variants, the normalised expression of genes within ±1 Mb distance to them were analysed as phenotype. In other words, the analysis focussed on _cis_ eQTL genes for these large-effect variants were analysed. When GWAS results of gene expression were obtained (_cis_ eQTL), the effect allele was mapped to the ancestral allele to determine the effects of MAs. For quantifying the number of eQTL for each effect direction of MAs, only the SNPs with the smallest $p$ value were considered.

**Reporting summary**. Further information on research design is available in the Nature Research Reporting Summary linked to this article.

## Data availability
Our predictions of cattle ancestral alleles for those 42,573,455 sites have been made publicly available at https://melbourne.figshare.com/articles/dataset/The_assignment_of_cattle_ancestral_alleles/13546472. Multiple alignment data used to determine cattle ancestral alleles are publicly available via Ensembl EPO pipeline (http://asia.ensembl.org/info/genome/compara/multiple_genome_alignments.html). Australian farmers and DataGene Australia (http://www.datagene.com.au/) are owners and custodians of the raw phenotype and genotype data of Australian dairy animals. Access to these data for research requires permission from DataGene under a Data Use Agreement. The DNA sequence data as part of the 1000 Bull Genomes Consortium[23–25] are available to consortium members and the membership is open. Sequence data of 1832 samples from the 1000 Bull Genome Project have been made publicly available at https://www.ebi.ac.uk/eva/?eva-study=PRJEB42783. The gene expression data are publicly available (NCBI SRA SRP111067). In addition: (1) The summary data of the effect direction and effect category of those 7.9 M sequence variants for which the ancestral alleles can be assigned are published at https://melbourne.figshare.com/articles/dataset/Summary_of_the_direction_and_category_of_effects_of_analysed_variants_on_37_traits/15170916; (2) The allele frequency of mutant alleles of those 7.9 M sequence variants for which the ancestral alleles can be assigned for the Holstein and Jersey cattle from the 1000 Bull Genome Project are published at https://melbourne.figshare.com/articles/dataset/Allele_frequency_of_mutant_alleles_in_Holstein_and_Jersey_from_the_1000_Bull_Genomes/15170922; (3) The coordinates of conserved sites analysed in the manuscript are published at https://melbourne.figshare.com/articles/dataset/Coordinates_of_cattle_conserved_sites/15170928. Other supporting data are shown in the Supplementary Materials of the current manuscript. Source data of figures can be found at https://melbourne.figshare.com/articles/dataset/Source_data_for_Mutant_alleles_differentially_shape_fitness_and_other_complex_traits_in_cattle/16869957.

## Code availability
The probability of ancestral allele assignment was estimated using the software estsfs published by Keightley et al.[28]. The linear mixed model used GCTA[50] and MTG2[51]. The Bayesian analysis used BayesR[61]. The R code of estimating heterozygosity across variants at conserved sites is at https://github.com/rxiangr/Genome-wide-heterozygosity.

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

## Acknowledgements
Australian Research Council's Discovery Projects (DP160101056 and DP200100499) supported R.X. and M.E.G. DairyBio, a joint venture project between Agriculture Victoria (Melbourne, Australia), Dairy Australia (Melbourne, Australia) and the Gardiner Foundation (Melbourne, Australia), funded computing resources used in the analysis. The authors also thank the University of Melbourne, Australia for supporting this research. No funding bodies participated in the design of the study nor analysis or interpretation of data nor writing of the manuscript. DataGene provided access to the reference data used in this study and the 1000 Bull Genomes consortium provided access to cattle sequence data. We thank Gert Nieuwhof and Kon Konstantinov (DataGene) for the preparation and provision of data. We thank Professor Hans D. Daetwyler for assisting with the access to the 1000 Bull Genomes data. We thank Professor Naomi Wray for a critical read of the manuscript.

## Author contributions
M.E.G. and R.X. conceived the study. R.X. performed all analyses. E.J.B. contributed to the BayesR analysis. I.M.M. assisted with data curation. S.B., C.J.V.J. and A.J.C. contributed to the imputation of sequence variants. R.X. and M.E.G. wrote the paper. R.X., M.E.G., E.J.B. and I.M.M. revised the paper. All authors read and approved the final manuscript.

## Competing interests
The authors declare no competing interests.
