## [Peer Review File · Communications Biology]

Reviewers' comments:

Reviewer #1 (Remarks to the Author):

Xiang et al. reported the effect of conserved and mutant alleles on the phenotypic traits in cattle. The results are interesting, but several major concerns should be considered before acceptance for publication.

1. The inference of ancestral alleles. The authors have used the species of sheep, yak and camel in inferring the ancestral alleles. What is the explanation for the utilization of these three species in the inference of ancestral alleles? How about goat, which is the same number of chromosomes as that of cattle? The correct choice of an outgroup in the inference of ancestral alleles is very important.

2. The accuracy of SNPs. It is very important to validate the accuracy of the SNP alleles called from sequences, especially when the depth is not high ($\times 10$). Since there are BeadChip alleles available here, it would be possible to compare the genotypes from the sequences and BeadChip SNPs.

3. The correction among the traits. There might be correction among the set of traits studied here. What is the impact of the correction on the results obtained?

4. In the GWAS, the factor of sex has not been considered. I think there must be impact from the sex difference.

5. It would be very important to validate the effect of alleles at the RNA or protein level. Also, what are the locations of the conserved or ancestral alleles and mutant alleles? What are their effect on the traits at the gene expression level? Is there difference between the origins of the ancestral alleles from parents?

6. I also found some paragraphs are too long, and these can be divided into shorter paragraphs.

Reviewer #2 (Remarks to the Author):

Initially, the paper had considerable intrigue for me. However, upon carefully scrutinising the title, reading the abstract and then the introduction a number of times, and carefully going through the Methods, I personally found the paper very frustrating. Perhaps that reflects my inadequate comprehension skills rather than that of the explanatory skills of the authors. The paper represents a considerable body of effort, and probably contains some useful findings from my perspective. However, the descriptions contain a lot of ambiguity in my mind, and many facts seem to be inconsistent in different parts of the paper.

The title suggests that "mutant alleles" have a different effect on complex traits relative to the effect they have on fitness traits. That is, the title introduces a classification of "traits" as being "complex" or as being involved with "fitness". It also suggests that "alleles" must have a classification as "mutant" or something else, as yet undefined. Unfortunately, the paper is not consistent in its terminology or use of this and other classifications. I presumed we would learn more from the abstract.

The abstract in line 1 introduces the concept of "classical mutations". It's not clear whether these are the same as "mutant alleles". In the second sentence, it introduces the concept of mutations with "polygenic effects". I would have thought polygenic effects refers to something controlled by more than one locus, typically many loci. So I am not sure what a mutation that has a "polygenic effect" means. Did the authors mean pleiotropic effects? The next sentence introduces the concept of alleles as being "mutant" and "ancestral". I presume they meant that alleles are ancestral "or" more recent mutations? But it is hard to be sure, at this stage of reading the paper. At any rate,

we are told that 8 m sequence variants were classified as presumably ancestral OR mutant. Since this would only make sense for polymorphic loci, if these were bi-allelic loci, the 8m variants would imply 4 m loci. Unless "variant" does not mean variant in the sense of an alternative or reference allele, but is now referring to a loci. The remainder of the abstract makes no mention of the distinction between complex traits and fitness traits that I presumed "mutant" alleles differentially shape.

The abstract then introduces the concept of heterozygosity at conserved sites. It's not clear whether "conserved" sites are a subset of the presumably 8m loci, or refers to all those 8 m "variants" for which we were just told that "ancestral" alleles had been assigned. It then states a list of complex traits (fertility, stature and milk production) as being "associated" with fitness. Or did the authors mean that the conserved "sites" were associated with fitness, rather than the traits?

The abstract then introduces yet another classification. This one is to do with direction of the effect in relation to the frequency of the mutant allele. So have mutant alleles now been classified as having positive and negative effects, in addition to be cross classified in terms of allele frequency?

Finally, the abstract introduces the concept that there are two classes of "genomic sites" (not alleles, or variants), those conserved across vertebrates and those others that are "subject to less long-term selection".

I thought the introduction might clarify my inability to understand what was meant by the title and the abstract. Line 27 defines "classical mutations, with a large effect on phenotype". I am not sure if this means all classical mutations have a large effect, or among classical mutations, some have a large effect. I am not provided a better understanding or what I meant by a "classical mutation". Does it mean a mutant that has been segregating for a long time – perhaps an ancestral mutant? The authors don't use the term "wild type" which I thought might be what they meant by ancestral alleles. Line 29 now informs me that there are "mutations of small or polygenic effect". Do the authors mean "small, polygenic effect" – ie "and" rather than "or"? In line 51 the authors introduce the concept of "new mutations". They state traits linked to fitness show inbreeding depression and heterosis, caused by "directional dominance".

We are then told that a method will be introduced to "test for directional dominance" by estimating the effects of heterozygosity, and this will be used to identify "traits that are associated with fitness". It is not clear if this is something to be done at a locus level, or a trait level.

We are then introduced to the concept that "genomic sites that are conserved across many species are likely to be functional". Further, that they will "use yak, sheep and camel" (presumably the many species referred to in the previous sentence), to identify "ancestral alleles"....."at 8m sequence variant sites". So now I have the impression there are 8 m conserved sites. But in the next sentence I am told the effect of heterozygosity is estimated at "both conserved sites and all genomic sites". So perhaps only some of the 8 m variants are at conserved sites. In which case what is an "ancestral allele"? Lastly, the analysis is going to be extended to genomes of ancient and modern cattle from the 1000 bull genomes database. I was not aware there were ancient genomes in the 1000 bull genomes database. And I don't understand how the "analysis can be extended to additional genomes" when the analysis of directional dominance presumably requires phenotypes – which the authors have for 37 traits on 4 breeds of "modern cattle".

Perhaps the results section will make all this clearer to the reader. But now we are told about results of almost 16 m "sites". I had thought there were 8m. Then further, at line 99, we are told the authors assigned ancestral alleles to 19 m sites. Then at line 14 we are told GWAS was done for 16 m sites. Then "for about 7m for which the ancestral allele was assigned"...does this mean 9 m did not have ancestral allele assigned? Or were discarded for some other reason? Then "the same comparison was performed for the 317,000 conserved sites". Are they not a subset of the sites for which ancestral alleles were assigned?

Then we are given statements like "However, there are many sites where the direction of selection varies between species and between breeds within species and where the effect direction of

mutations can be either positive or negative." This doesn't seem like any surprise.

Without going into all the details, the Methods section also contained inadequacies in terms of my ability to comprehend what was done. At line 407 breed was fitted in the model, but a polygenic effect based on a genomic relationship matrix was also fitted, based on all variants. Presumably the effect of breed is genetic, and the polygenic random effects should capture this since the G matrix was built with all variants. Line 411 indicates G=genomic relatedness matrix based on all variants – but there are many ways of constructing a G matrix – particularly across breeds. The authors don't bother to tell us about the distributional assumptions for the error term in the model. I would not be able to exactly repeat the analyses described in the methods as the details do not contain enough precision.

The BayesR model on line 427 is vague. We are told W was a design matrix for genotypes. This implies to me that W accounted for genotype classes. The vector v was the vector of variant effects. There was a design matrix for fixed effects, including breed. Again, I would have thought breed would be excluded since it's a genetic effect. Further, the vector of fixed effects "including breeds" suggests that some other fixed effects were fitted. What were those? Then we are told about the effect b for each variant – does this mean a scalar effect for each locus, rather than a class genotype effect? So was the W matrix really a matrix of covariates rather than class effects? But I thought variant effects were in v, not in the vector b of fixed effects?

Line 477 refers to computing inbreeding depression. I don't understand how the effect of inbreeding on a trait, which is what inbreeding depression is, can be inferred from heterozygosity and allele frequencies without phenotypes. Or is "inbreeding depression" really about homozygosity?

Line 481 refers to estimation of H_j not being orthogonal to additive components. I don't understand orthogonal in this context.

Line 548 of Data and Code availability states that "Datagene are custodians of the raw phenotype and genotype data". It does not state whether the exact data used in this paper are stored in an exactly reproducible form or not, or whether they would be made available to other parties.

Reviewer #3 (Remarks to the Author):

This is an interesting analysis of the impact of mutant (or derived) alleles on traits in cattle. The authors make use of a massive data set to assess the effect of novel alleles on traits which have been quantified in a multiple breed of cattle. Aside for a few details, I found the analysis and conclusions to be sound (aside for a few details below). In general I think the paper is interesting to a broad readership of the journal and the paper should be accepted with a few revisions and if the data is made publicly available.

Having said that, and although I found the results intriguing, I am not sure about their relevance in a wider (evolutionary) biological context. In particular, it is difficult to see how the traits analysed in this study can be linked to actual fitness. This is a major drawback of the study as it makes it difficult to address the central question posed by the authors. The fact that some derived alleles increase some trait values, while others decrease some other traits values is not all that surprising - unless the value of these traits can be linked to fitness though, the relevance of the findings are difficult to assess. I feel that the introduction and discussion could expand about more around this topic : "When practicing genomic selection, which is widely used in animal breeding, it would be an advantage to know a priori whether mutations are more likely to increase or decrease traits of interest." - which I think is more interesting.

The issue of the relevance of the finding is made more difficult to assess as the introduction (and abstract) is in some places quite confusing. For example the authors are saying that large effect mutations are "classical" mutations, then ask the question: "Do these mutations show the same characteristics as classical mutations?" and then seem to answer their own question by: "Mutations

with small effect on fitness tend to be deleterious" This is quite a confusing paragraph. Also, aren't most mutations: small effect and slightly deleterious?

The introduction then jumps onto the concept of directional dominance, without introducing it or linking it to the previous section. We are also told that: "Traits that are related to fitness typically show inbreeding depression and heterosis caused by directional dominance", without a reference or any rationale. I think this could be expanded to give the reader a bit more biological context for the interpretation of this rather complex set of results.

I am also concerned about the availability of the data. It is unclear to me whether the whole 1000 bull genome project data used here has been made publicly available. The authors mention in their method section that they used 98 samples from the 1,000 bull genome projects but in the main text (line 187) they say: "1,720 ancient and modern cattle from the 1000 Bull Genomes Project". Which one is it? Are all these 1,720 individuals available on SRA/ENA? What about the genotypic/phenotypic data for all the individuals, maybe I missed something but I could only find some of the phenotypic data on the datagene.com.au website? The authors should upload the phenotypic and genotype data to a long term repository (e.g. dryad?) so that the exact same dataset can be accessed.

Other comments:

The term mutant allele is not very common in the field of population genetics. Derived allele is a lot more common.

Line 34: Cattle are artiodactyl.

Line 55: The sentence that starts with "Genomics" does not follow the previous one.

Line 60: "at 8M sequence variant sites on 37 traits of 4 breeds of over 113k modern cattle." This is very confusing, the variants on traits? Consider rephrasing.

I am a bit concerned about the way that the author discretized the phastcons score. 0.9 seems a bit arbitrary? Is there any justification for this?

The font of the figures are too small to read.

How did the author call aligned reads and called SNPs in the 98 (or 1.7k individuals) from the 1,000 bull genome project? How did they treat the ancient DNA data?

Supplementary Table 2 should include individual accession numbers.

Response to referees COMMSBIO-21-1004-T:

‘Mutant alleles differentially shape cattle complex traits and fitness’

We thank the reviewers for their time and effort in providing comments and suggestions on our manuscript. Please find our point-to-point response to reviewers’ comments in blue text in the following.

Reviewers' comments:

Reviewer #1 (Remarks to the Author):

Xiang et al. reported the effect of conserved and mutant alleles on the phenotypic traits in cattle. The results are interesting, but several major concerns should be considered before acceptance for publication.

Author response: We thank the positive comments from the reviewer. Please see our detailed responses to your concerns in the following text.

1. The inference of ancestral alleles. The authors have used the species of sheep, yak and camel in inferring the ancestral alleles. What is the explanation for the utilization of these three species in the inference of ancestral alleles? How about goat, which is the same number of chromosomes as that of cattle? The correct choice of an outgroup in the inference of ancestral alleles is very important.

Author response: we apologize for not clarifying this methodological detail. We added the following section to the methods: ‘According to the evolutionary relationships reported previously [1], among ruminants, yak is an outgroup species closely related to cattle, while sheep is less closely related to cattle than yak. Goat is equivalent to sheep in its relationship to cattle, but we chose sheep in the current study.’ As stated in the 2nd paragraph of Results, the ancestral alleles generated from the current study have a strong agreement with a previous study (<https://pubmed.ncbi.nlm.nih.gov/24862839/>).

2. The accuracy of SNPs. It is very important to validate the accuracy of the SNP alleles called from sequences, especially when the depth is not high (X10). Since there are BeadChip alleles are available here, it would be possible to compare the genotypes from the sequences and BeadChip SNPs.

Author response: we agree with the reviewer that this is important. Therefore, the following descriptions have been added/revised to the section of '**Genotype data for association analysis.**' in Methods: 'Prior to sequence imputation, the HD genotypes were converted to forward sequence format. Then, all HD genotypes were imputed to sequence using Minimac3 with Eagle (v2) to pre-phase genotypes ([2, 3]). The reference set for imputation included sequences of 3090 *Bos taurus* animals from Run7 of the 1000 Bull Genomes Project [4] aligned to the ARS-UCD1.2 reference bovine genome (https://www.ncbi.nlm.nih.gov/assembly/GCF_002263795.1/) [5, 6]. The accuracy of the sequence data for individual animals in the 1000 Bull Genomes Project is routinely checked against their own high-density SNP array genotypes and the concordance has been above 95% [7]. The empirical accuracy of imputation to sequence using the 1000 Bull Genomes project has been routinely tested for dairy breeds: for example, in Holsteins the average correlation between imputed and real sequence variants was 0.92 to 0.95 using Run5 of the 1000 Bull Genomes project (N= 1577)[8]. Therefore, we believe our imputed data is more accurate: first because the number of reference animals has almost doubled and second because in our study we impose a Minimac3 R^2 filter to remove poorly imputed variants. A Minimac3 R^2 threshold of 0.4 was used because our in-house tests demonstrate that this is approximately equivalent to an empirical imputation accuracy (correlation) of 0.85.'

3. The correction among the traits. There might be correction among the set of traits studied here. What is the impact of the correction on the results obtained?

Author response: there are two places in the manuscript where we describe trait correction. One place is 'Daughter trait deviations were the average trait deviations of a bull's daughters and all phenotypes were **pre-corrected** for known fixed effects, with processing done by DataGene.' The correction of fixed effects is standard procedure for DataGene to process data for analysis and publication of the Australian dairy cattle evaluations. We received the data already corrected for the standard effects of herd, year, season and cow age. The other place that 'correction' is mentioned: 'we have **corrected** the trait direction so that larger

values of Fert, Ease, Temp, MSpeed and Like meant increased fertility performance (calving frequency), labour ease, docility, milking speed and the overall preference as a dairy cow'. This correction simply reverses the trait value direction and has no impact on the results other than to help the reader understand the trait definitions. We have added this explanation to the section of 'Animals and phenotypes for variant-trait association analysis' in Methods.

4. In the GWAS, the factor of sex has not been considered. I think there must be impact from the sex difference.

Author response: In this study, the 'Phenotype' of bulls is based on the data from their female relatives ('Daughter trait deviations were the average trait deviations of a bull's daughters' in the section of **Animals and phenotypes for variant-trait association analysis of Methods**). Therefore, in fact all data is on female performance and therefore, no sex factor is involved.

5. It would be very important to validate the effect of alleles at the RNA or protein level. Also, what are the locations of the conserved or ancestral alleles and mutant alleles? What are their effect on the traits at the gene expression level? Is there difference between the origins of the ancestral alleles from parents?

Author response: The coordinates of ancestral alleles used in the study have been made publicly available: <https://figshare.com/s/dd5985b76a413b56106b>. We have also uploaded the coordinates of conserved sites and these data can be accessed via link <https://figshare.com/s/df9d3662f8f7fb8e72da>.

The reviewer proposed to use the effects of alleles on intermediate traits, such as the expression of genes to validate the effects of alleles on complex traits. We agree with the reviewer that the analysis of allele effects on intermediate traits is interesting. However, such analysis cannot validate our results. It is known that effects of many alleles (or SNPs) on gene expression does not translate to allele effects on complex traits (e.g., [9, 10]) and many SNPs affect complex traits with no effect on gene expression. In other words, whether an allele increases or decreases the expression of a gene is not necessarily related to its effects on traits like milk yield or height.

Further, we would like to stress that our study has already included 4 validations of results: 1) comparison of the call of ancestral alleles with previously generated data, 2) analysis of allele effects using GWAS validated by allele effects using BayesR [Figure 2], 3) using alleles that

are significant and in the same effect direction in independent bull and cow datasets [1-3 paragraph of the section ‘Biases in trait effects between ancestral and mutant alleles’] and 4) validate allele frequency distribution discovered in the Australian cattle data in the external 1000 Bull Genomes Project data [Figure 3]. All 4 sets of validations supported the robustness of our results and therefore, the conclusion.

Nevertheless, to address the comment from the reviewer, we conducted the following additional analysis. We selected those top GWAS variants with large effects ($p < 5e-8$) on protein yield, fat yield, milk yield, protein % and fat %. For top variants related to a trait, we divide them into two subsets based on whether the mutant allele decreases (5 sets of variants) or increases (another 5 sets of variants) each trait. We then test if the mutant allele which decreases/increases the trait also tends to decrease/increase the expression of its cis eQTL genes in milk cells [11]. This was done in milk cells because it is biologically mostly related to milk production traits. We found 4 out of 5 sets of variants, where the mutant allele decreased the trait, the mutant allele of these variants tended to decrease the expression of cis eQTL genes. For another 4 out of 5 sets of variants where mutant allele increased the trait, the mutant allele of these variants tended to increase the expression of cis eQTL genes. These results support that the effects of mutant alleles on traits are not random and appear to be related to their effects on gene expression. We have added those results to the 4th paragraph in the ‘*Biases in trait effects between ancestral and mutant alleles*’ section of Results and Supplementary Table 3.

While we think the question raised by the reviewer ‘Is there difference between the origins of the ancestral alleles from parents?’ is interesting, this is out of the scope of the current manuscript. We prefer to pursue this idea in future projects.

6. I also found some paragraphs are too long, and these can be divided into shorter paragraphs.

Author response: We thank the suggestion from the reviewer. In the revised manuscript, the 4th paragraph of the introduction is divided into two paragraphs. The first 2 paragraphs of the section of ‘Biases in trait effects between ancestral and mutant alleles’ in Results is divided into 4 paragraphs. We think the size of other paragraphs are ok.

Reviewer #2 (Remarks to the Author):

Initially, the paper had considerable intrigue for me. However, upon carefully scrutinising the title, reading the abstract and then the introduction a number of times, and carefully going through the Methods, I personally found the paper very frustrating. Perhaps that reflects my inadequate comprehension skills rather than that of the explanatory skills of the authors. The paper represents a considerable body of effort, and probably contains some useful findings from my perspective. However, the descriptions contain a lot of ambiguity in my mind, and many facts seem to be inconsistent in different parts of the paper.

Author response: we thank the reviewer for acknowledging the effort in the manuscript. We apologize for the descriptions of terms. In the revised manuscript we have gone through and tried to clarify as much as possible. Please see our detailed responses in the following text.

The title suggests that “mutant alleles” have a different effect on complex traits relative to the effect they have on fitness traits. That is, the title introduces a classification of “traits” as being “complex” or as being involved with “fitness”. It also suggests that “alleles” must have a classification as “mutant” or something else, as yet undefined. Unfortunately, the paper is not consistent in its terminology or use of this and other classifications. I presumed we would learn more from the abstract.

Author response: we are trying to be succinct in title. We in general agree with the interpretation of the title by the reviewer, which may suggest that the title is ok. We therefore have kept the title. Fitness might be considered one of many complex traits.

The abstract in line 1 introduces the concept of “classical mutations”. It’s not clear whether these are the same as “mutant alleles”. In the second sentence, it introduces the concept of mutations with “polygenic effects”. I would have thought polygenic effects refers to something controlled by more than one locus, typically many loci. So I am not sure what a mutation that has a “polygenic effect” means. Did the authors mean pleiotropic effects? The next sentence introduces the concept of alleles as being “mutant” and “ancestral”. I presume they meant that alleles are ancestral “or” more recent mutations? But it is hard to be sure, at this stage of reading the paper. At any rate, we are told that 8 m sequence variants were classified as presumably ancestral OR mutant. Since this would only make sense for polymorphic loci, if these were bi-allelic loci, the 8m variants would imply 4 m loci. Unless “variant” does not mean variant in the sense of an alternative or reference allele, but is now

referring to a loci. The remainder of the abstract makes no mention of the distinction between complex traits and fitness traits that I presumed “mutant” alleles differentially shape.

Author response: We apologize for the errors and have revised our abstract.

To improve the consistency we have changed ‘classical mutations’ to ‘classical mutant alleles (MAs)’ and in the 1st sentence of the abstract we provide a definition of classic MAs. It now reads ‘Classical mutant alleles (MAs), with large effects on phenotype, tend to be deleterious to traits and fitness.’.

We agree with the reviewer that the use of the term ‘polygenic effects’ is inaccurate.

Therefore, we have changed it to ‘small effects’.

To avoid confusion, we have removed the term ‘ancestral’ here because as the reviewer said, it can only be 1 of them.

To our knowledge, a variant or SNP cannot be classified as ancestral or mutant, only an allele for a variant can be classified as such. We therefore think the revised sentence ‘we infer MAs for 8 million sequence variants in 113k cattle...’ is ok.

We did make a distinction between complex traits and fitness in the abstract: ‘Heterozygosity at sites conserved across 100 vertebrates increase fertility, stature, and milk production, positively associating these traits with fitness.’ This means that not all 37 traits are related to fitness.

The abstract then introduces the concept of heterozygosity at conserved sites. It’s not clear whether “conserved” sites are a subset of the presumably 8m loci, or refers to all those 8 m “variants” for which we were just told that “ancestral” alleles had been assigned. It then states a list of complex traits (fertility, stature and milk production) as being “associated” with fitness. Or did the authors mean that the conserved “sites” were associated with fitness, rather than the traits?

Author response: we have added ‘for variants’ after ‘Heterozygosity’ to link this sentence to whats described before, i.e., they are a subset of 8M variants. We also added ‘genomic’ to ‘sites’ to indicate these are positions in the genome.

The abstract then introduces yet another classification. This one is to do with direction of the effect in relation to the frequency of the mutant allele. So have mutant alleles now been classified as having positive and negative effects, in addition to be cross classified in terms of allele frequency?

Author response: We agree with the reviewer in interpreting this section. An allele can have positive (increase) or negative (decrease) effects on a trait and can also have low or high frequency in the population.

Finally, the abstract introduces the concept that there are two classes of “genomic sites” (not alleles, or variants), those conserved across vertebrates and those others that are “subject to less long-term selection”.

Author response: in addressing previous comments of the reviewer (genomic sites conserved across 100 vertebrates...), the term genomic sites are introduced. We hope this can address the concerns of the reviewer.

I thought the introduction might clarify my inability to understand what was meant by the title and the abstract. Line 27 defines “classical mutations, with a large effect on phenotype”. I am not sure if this means all classical mutations have a large effect, or among classical mutations, some have a large effect. I am not provided a better understanding or what I meant by a “classical mutation”. Does it mean a mutant that has been segregating for a long time – perhaps an ancestral mutant? The authors don’t use the term “wild type” which I thought might be what they meant by ancestral alleles. Line 29 now informs me that there are “mutations of small or polygenic effect”. Do the authors mean “small, polygenic effect” – ie “and” rather than “or”? In line 51 the authors introduce the concept of “new mutations”. They state traits linked to fitness show inbreeding depression and heterosis, caused by “directional dominance”.

Author response: The complete sentence that the reviewer refers to as ‘Classical mutations, with a large effect on phenotype, tend to decrease fitness, decrease fitness-related traits and be partially recessive¹⁻³.’ This means that the classical mutations described here are those ones with large effect and harmful to fitness, which are described in the first 3 references. To clarify this for the reviewer, we have added ‘(also see the 1st category of mutations defined in ³)’ to the end of this sentence.

We agree with the reviewer that the word ancestral is equivalent to wild type. However, we chose the word ancestral over wild type because we study cattle which is a domestic animal and ancestral alleles may be more appropriate.

We have removed the term 'polygenic' to avoid confusion.

We have removed the 'new' in the term 'new mutations' to avoid confusion.

We are then told that a method will be introduced to “test for directional dominance” by estimating the effects of heterozygosity, and this will be used to identify “traits that are associated with fitness”. It is not clear if this is something to be done at a locus level, or a trait level.

Author response: we have added 'from genomic sites' so that this sentence reads 'we introduce a method testing for directional dominance by estimating the effect of heterozygosity at genomic sites on traits of cattle...'. It is traits which either show directional dominance or not and it is traits that show inbreeding depression or heterosis. We have also added more descriptions to describe the purpose of testing for directional dominance to improve the readability at this section (the current 3rd paragraph of Introduction).

We are then introduced to the concept that “genomic sites that are conserved across many species are likely to be functional”. Further, that they will “use yak, sheep and camel” (presumably the many species referred to in the previous sentence), to identify “ancestral alleles”.....”at 8m sequence variant sites”. So now I have the impression there are 8 m conserved sites. But in the next sentence I am told the effect of heterozygosity is estimated at “both conserved sites and all genomic sites”. So perhaps only some of the 8 m variants are at conserved sites. In which case what is an “ancestral allele”? Lastly, the analysis is going to be extended to genomes of ancient and modern cattle from the 1000 bull genomes database. I was not aware there were ancient genomes in the 1000 bull genomes database. And I don't understand how the “analysis can be extended to additional genomes” when the analysis of directional dominance presumably requires phenotypes – which the authors have for 37 traits on 4 breeds of “modern cattle”.

Author response: the reviewer describes two sections located in two different paragraphs, with which we tried to use to avoid confusion. This section has been revised as 'A likely cause of directional dominance is that mutations tend to be deleterious and partially recessive.

However, not all sites in the genome affecting a trait may show this pattern. Our second objective was to test the hypothesis that sites, where the same allele has been conserved across vertebrate evolution, are the most likely to show directional dominance. Therefore, we compared directional dominance at conserved sites versus other polymorphic sites in this analysis.'

We agree with the interpretation of the reviewer that 'some of the 8 m variants are at conserved sites.' The ancestral allele at each site is the older allele. Usually at biallelic sites, such as SNPs, the derived allele has arisen by mutation from the ancestral allele. The ancestral allele is likely to be an allele carried by related species. Here we use yak, sheep and camel as related species. However, an allele that is shared by these 3 species is not necessarily shared across 100 species of vertebrates (conserved sites).

We apologize for ambiguities in describing the additional analysis in the 1000 Bull Genomes. This section now reads 'If mutant alleles decrease fitness we expect selection to reduce their allele frequency compared with mutant alleles that either have no effect or increase fitness. Therefore, we compare the allele frequency of mutant alleles that increase and decrease each trait. We expand the analysis of mutant allele frequency to additional breeds of ancient and modern cattle from the 1000 Bull Genomes database [4, 6], which provides validation of our results.'

Perhaps the results section will make all this clearer to the reader. But now we are told about results of almost 16 m "sites". I had thought there were 8m. Then further, at line 99, we are told the authors assigned ancestral alleles to 19 m sites. Then at line 14 we are told GWAS was done for 16 m sites. Then "for about 7m for which the ancestral allele was assigned"...does this mean 9 m did not have ancestral allele assigned? Or were discarded for some other reason? Then "the same comparison was performed for the 317,000 conserved sites". Are they not a subset of the sites for which ancestral alleles were assigned?

Author response: we apologize for the confusion. In the revised version, we provide the following text in the 1st paragraph of the results: 'In total, there were 16,035,443 imputed sequence variants (at 16,035,443 genomic sites) with imputation accuracy $R^2 > 0.4$ and the minor allele frequency (MAF) > 0.005 available for variant-trait association analysis. A subset of these sequence variants that could be assigned with ancestral alleles was used for analyses related to mutant alleles (described later). For the analysis of the effect of heterozygosity, we fit the average heterozygosity of sequence variants at 317,279 genomic

sites conserved across 100 vertebrates (H'_{cons_j}) and heterozygosity from variants at the other 15,718,164 sites ($H'_{non-cons_j}$) simultaneously (see Methods).'

For conserved sites, we have specified the number in the 3rd paragraph of results: 'The same comparison was also performed for variants at 202,530 out of 317,279 conserved sites that ancestral alleles could be assigned.'

Then we are given statements like "However, there are many sites where the direction of selection varies between species and between breeds within species and where the effect direction of mutations can be either positive or negative." This doesn't seem like any surprise.

Author response: the reviewer refers to the very last sentence of the conclusion. We agree with the reviewer that this statement is not surprising. Therefore, we have extensively revised the conclusion: 'In conclusion, our results support a new hypothesis which provides a new picture of the effects of mutation and selection on mammalian complex traits. Directional dominance, which causes heterosis and inbreeding depression, is characteristic of loci where the mutations decrease the trait and fitness and this pattern has been consistent over the evolution of vertebrates. More recent selection, although not causing directional dominance, leads to a bias in the direction of mutation because the mutation results in an allele which is less fit than the ancestral allele and tends to affect a complex trait in a consistent direction. This hypothesis, if supported by future research, adds to our understanding of the evolution of complex traits and has practical value in the artificial selection of livestock and crops.'

Without going into all the details, the Methods section also contained inadequacies in terms of my ability to comprehend what was done. At line 407 breed was fitted in the model, but a polygenic effect based on a genomic relationship matrix was also fitted, based on all variants. Presumably the effect of breed is genetic, and the polygenic random effects should capture this since the G matrix was built with all variants. Line 411 indicates G=genomic relatedness matrix based on all variants – but there are many ways of constructing a G matrix – particularly across breeds. The authors don't bother to tell us about the distributional assumptions for the error term in the model. I would not be able to exactly repeat the analyses described in the methods as the details do not contain enough precision.

Author response: we thank the careful consideration of the reviewer on our methods. We agree with the interpretation of the model by the reviewer. The sentence ‘The purpose of fitting breeds as fixed effects together with the GRM in the model is to have strong control of the population structure which may cause spurious associations between variants and phenotype.’ Was added to this section in Methods.

We apologize for the ambiguity in describing the model and this section has been updated: ‘The above mentioned traits were analysed one trait at a time independently in each sex with linear mixed models using GCTA [12]:

$$\mathbf{y} = \mathbf{mean} + \mathbf{breed} + \mathbf{bx} + \mathbf{a} + \mathbf{error} \quad (\text{equation 1})$$

where \mathbf{y} = vector of phenotypes for bulls or cows, **breed** = three breeds for bulls, Holstein, Jersey and Australian Red and four breeds for cows (Holstein, Jersey, Australian Red and MIX); **bx** = regression coefficient b on variant genotypes \mathbf{x} ; **a** = random polygenic effects $\sim N(0, \mathbf{G}\sigma_g^2)$ where \mathbf{G} = genomic relatedness matrix based on all variants and σ_g^2 = random polygenic variance; **error** = random residual effects $\sim N(0, \mathbf{I}\sigma_e^2)$, where \mathbf{I} = the identity matrix and σ_e^2 the residual variance.’

We apologize for the ambiguity in describing the construction of GRM, but it followed the default procedure of the software used (GCTA). We have also specified this in this section: The construction of GRM followed the default setting (--make-grm) in GCTA[12]: (<https://cnsgenomics.com/software/gcta/#MakingaGRM>).

The BayesR model on line 427 is vague. We are told W was a design matrix for genotypes. This implies to me that W accounted for genotype classes. The vector v was the vector of variant effects. There was a design matrix for fixed effects, including breed. Again, I would have thought breed would be excluded since it’s a genetic effect. Further, the vector of fixed effects “including breeds” suggests that some other fixed effects were fitted. What were those? Then we are told about the effect b for each variant – does this mean a scalar effect for each locus, rather than a class genotype effect? So was the W matrix really a matrix of covariates rather than class effects? But I thought variant effects were in v , not in the vector b of fixed effects?

Author response: We apologize for these ambiguities and there are no other fixed effects than breeds. We have replaced ‘including’ with ‘of’ so the sentence reads ‘...b was the vector of fixed effects of breeds;’. Again, we fit breed as fixed effects to control spurious associations as done in GWAS.

We apologize for the typo here it should be ‘v’ instead of ‘b’ to be the vector of variant effects. We have corrected this typo.

Line 477 refers to computing inbreeding depression. I don’t understand how the effect of inbreeding on a trait, which is what inbreeding depression is, can be inferred from heterozygosity and allele frequencies without phenotypes. Or is “inbreeding depression” really about homozygosity?

Author response: the reviewer is right that the calculation is about homozygosity or inbreeding coefficient. We have replaced ‘depression’ with ‘coefficient’ so the sentence reads ‘Note that such calculation of H_j can also be used to derive inbreeding coefficient, where $I_j = (\sum_i^N 2p_i q_i) \times H_j$. I_j was the inbreeding coefficient for the j^{th} individual.’

Line 481 refers to estimation of H_j not being orthologous to additive components. I don’t understand orthologous in this context.

Author response: we have modified this section to clarify this: ‘In equation 3, however, due to the non-zero correlation between x_A and z_D under Hardy-Weinberg equilibrium (HWE), the estimation of a and d is not independent, i.e., $cov(x_A, z_D) = 2p(1 - p)(1 - 2p) \neq 0$ under HWE. This then resulted in the estimation of H_j not being independent of the additive components.’

Line 548 of Data and Code availability states that “Datagene are custodians of the raw phenotype and genotype data”. It does not state whether the exact data used in this paper are stored in an exactly reproducible form or not, or whether they would be made available to other parties.

Author response: The raw genotype and phenotype data of Australian dairy cattle are third-party data and are the private property of Australian farmers and Datagene. As we don’t own the data, we don’t have the right to publish the data. Access to these data requires permission

from Datagene under a Data Use Agreement. This applies to any researchers who want to analyse their data, including ourselves.

Therefore, we have revised this section, including the description of newly added summary data, to clarify this: ‘Our predictions of cattle ancestral alleles for those 42,573,455 sites have been made publicly available at: <https://figshare.com/s/dd5985b76a413b56106b>. Multiple alignment data used to determine cattle ancestral alleles are publicly available via Ensembl EPO pipeline

(http://asia.ensembl.org/info/genome/compara/multiple_genome_alignments.html).

Australian farmers and DataGene Australia (<http://www.datagene.com.au/>) are owners and custodians of the raw phenotype and genotype data of Australian dairy animals. Access to these data for research requires permission from DataGene under a Data Use Agreement. The DNA sequence data as part of the 1000 Bull Genomes Consortium [4, 6, 13] are available to consortium members and the membership is open. Sequence data of 1832 samples from the 1000 Bull Genome Project have been made publicly available at:

<https://www.ebi.ac.uk/eva/?eva-study=PRJEB42783>. The gene expression data is publically available (NCBI SRA SRP111067). In addition: 1. The summary data of the effect direction and effect category of those 7.9M sequence variants for which the ancestral alleles can be assigned is published at <https://figshare.com/s/ef020d948523c31c0e67>; 2. The allele frequency of mutant alleles of those 7.9M sequence variants for which the ancestral alleles can be assigned for the Holstein and Jersey cattle from the 1000 Bull Genome Project is published at <https://figshare.com/s/20154b1d8e60e012e532>; 3. The coordinates of conserved sites analysed in the manuscript is published at: <https://figshare.com/s/df9d3662f8f7fb8e72da>. Other supporting data are shown in the supplementary materials of the current manuscript.’

Reviewer #3 (Remarks to the Author):

This is an interesting analysis of the impact of mutant (or derived) alleles on traits in cattle. The authors make use of a massive data set to assess the effect of novel alleles on traits which have been quantified in a multiple breed of cattle. Aside for a few details, I found the analysis and conclusions to be sound (aside for a few details below). In general I think the paper is interesting to a broad readership of the journal and the paper should be accepted with a few revisions and if the data is made publicly available.

Author response: we are grateful to the positive comments from the reviewer.

Having said that, and although I found the results intriguing, I am not sure about their relevance in a wider (evolutionary) biological context. In particular, it is difficult to see how the traits analysed in this study can be linked to actual fitness. This is a major drawback of the study as it makes it difficult to address the central question posed by the authors. The fact that some derived alleles increase some trait values, while others decrease some other traits values is not all that surprising - unless the value of these traits can be linked to fitness though, the relevance of the findings are difficult to assess. I feel that the introduction and discussion could expand about more around this topic : “When practicing genomic selection, which is widely used in animal breeding, it would be an advantage to know a priori whether mutations are more likely to increase or decrease traits of interest.” - which I think is more interesting.

Author response: we agree with the reviewer that it is difficult to link traits to fitness.

However, traits related to fitness typically show directional dominance causing inbreeding depression and heterosis. We have used this fact and so have classified traits showing directional dominance as ‘fitness traits’. This interpretation is supported by the finding that it is sites that have been conserved across 100 vertebrate species that are responsible for the directional dominance. Such long-term conservation must be due to consistent selection for one allele. For many complex traits (e.g. fat concentration in milk) it is unlikely that they are neutral (e.g. fat is costly to the female and beneficial to the infant). It is often assumed that such traits are subject to stabilising selection where an intermediate value leads to the highest fitness. Models of the evolution of such traits usually assume the mutation is equally likely to increase or decrease the trait. However, we find that for some traits (e.g. fat concentration in milk), mutation more often decreases the trait and that these mutant alleles are selected against. The simplest explanation of these results is that at the sites responsible for this, selection has been for alleles that increase the trait. That is, these sites in the genome have had a long-term effect on fitness so that the most favourable allele is the ancestral allele. However, traits such as fat concentration do not show heterosis or inbreeding depression. Thus we hypothesise two classes of genomic sites and traits related to fitness: those under extremely long-term consistent selection show directional dominance, while those with consistent selection but not of such a long-term nature show a bias in the direction of mutation. In the manuscript we have attempted to do this by associating heterozygosity from sites conserved across 100 vertebrates with traits. We found that protein, fat and milk yield, stature, fertility and survival are all positively related to fitness (Figure 1).

To emphasise this point in the results the following sections are modified:

In the 1st paragraph of Results, we modified the following section: ‘For all these traits, *heterozygosity at other sites* ($H'_{non-cons_j}$) was not significant when fitted together with H'_{cons_j} . This directional dominance implies that milk production, fertility, survival and stature show inbreeding depression and heterosis and therefore we classify them as fitness-related traits and this directional dominance for these traits is exclusively explained by genomic sites conserved across vertebrates.’

We changed the legend title of Figure 1 to ‘**Figure 1.** Directional dominance at conserved sites (H') for traits of 104k cows.’

In the 5th paragraph of ‘*Biases in trait effects between ancestral and mutant alleles*’: ‘In our study, MAs consistently showed biases towards decreasing protein and fat concentration (Figure 2 and Supplementary Figure 3,4), docility and stature, and towards increasing somatic cell count (an indicator of mastitis) and gestation length. **Among these traits only stature showed a significant effect of heterozygosity.** For milk yield and protein yield, **both of which were classified as fitness-related traits (Figure 1)**, the bias in the direction of MA depends on the size of the MA effect.’

In the 6th paragraph of ‘*Biases in trait effects between ancestral and mutant alleles*’: ‘Also, there was a slight majority of small-effect MAs which tended to increase fertility and survival, **both of which were positively related to fitness (Figure 1).**’

In the extensively revised Discussion and conclusion, we have extended the discussion regarding the directional dominance, fitness, and mutations and have tried to stress the novelty of the study. For example, the new conclusion now reads: ‘In conclusion, our results support a new hypothesis which provides a new picture of the effects of mutation and selection on mammalian complex traits. Directional dominance, which causes heterosis and inbreeding depression, is characteristic of loci where the mutations decrease the trait and fitness and this pattern has been consistent over the evolution of vertebrates. Less long-term selection, although not causing directional dominance, leads to a bias in the direction of mutation because the mutation results in an allele which is less fit than the ancestral allele and tends to affect a complex trait in a consistent direction. This hypothesis, if supported by future research, adds to our understanding of the evolution of complex traits and has practical value in the artificial selection of livestock and crops.’

We agree with the reviewer that the link between the direction of the effects of mutations and animal breeding is interesting. Therefore, we added a paragraph to the Discussion on this point (currently the 2nd last paragraph of Discussion before conclusion): ‘Genomic selection [14], used in the breeding of livestock and crops, estimates the genetic value of individuals for traits of interest from the alleles they carry at genetic markers such as SNPs. The equation predicting genetic value uses the effect of each SNP on the trait estimated in a training population. The best methods treat the SNP effects as random variables drawn from a prior distribution. To date it has been assumed that the effects of a mutation are equally likely to be positive or negative on the trait but, if it was known that one direction of effect was more likely, this could be built into the prior distribution resulting in an increase in the accuracy with which genetic value is predicted’.

The issue of the relevance of the finding is made more difficult to assess as the introduction (and abstract) is in some places quite confusing. For example the authors are saying that large effect mutations are "classical" mutations, then ask the question: "Do these mutations show the same characteristics as classical mutations?" and then seem to answer their own question by: "Mutations with small effect on fitness tend to be deleterious" This is quite a confusing paragraph. Also, aren't most mutations: small effect and slightly deleterious?

Author response: we apologize for these ambiguities and we have extensively revised/rewritten the introduction. For this particular section, the text has been revised as follows: ‘However, the majority of the genetic variance in complex traits is due to mutations of small effect. Do these **small-effect** mutations show the same characteristics as those classical **large-effect deleterious** mutations? A study in *E. coli* showed that mutations with small effect on fitness tend to be deleterious **to protein function** [15]. ... ’ we have also revised the abstract to clarify terms used in the manuscript.

The distribution of fitness effects of mutations is an ongoing research. However, a previous study (cited reference 3, Eyre-Walker, A. & Keightley, P. D. 2007) shows that there are many large-effect deleterious mutations (Figure 1 in this Eyre-Walker 2007). Also, most previous studies focused on flies or worms and our current study focuses on the effects on mutations on quantitative traits in a large animal. As stated in the 1st paragraph of introduction, ‘how mutations affect complex traits such as body size, health and fertility is unknown.’

The introduction then jumps onto the concept of directional dominance, without introducing it or linking it to the previous section. We are also told that: “Traits that are related to fitness typically show inbreeding depression and heterosis caused by directional dominance”, without a reference or any rationale. I think this could be expanded to give the reader a bit more biological context for the interpretation of this rather complex set of results.

Author response: we apologize for this ambiguity. This section has now been extensively revised/rewritten: ‘Traits that are related to fitness typically show inbreeding depression and heterosis caused by directional dominance. That is, fitness decreases with increased inbreeding due to increased homozygosity at loci carrying recessive deleterious alleles [16]. Conversely, fitness generally increases with heterozygosity [17]. Therefore, directional dominance can be used to link traits to fitness. Here, we introduce a method testing for directional dominance by estimating the effect of heterozygosity from genomic sites on traits of cattle and use this method to identify traits that are associated with fitness. Then, we classify traits showing directional dominance as ‘fitness-related traits’.

I am also concerned about the availability of the data. It is unclear to me whether the whole 1000 bull genome project data used here has been made publicly available. The authors mention in their method section that they used 98 samples from the 1,000 bull genome projects but in the main text (line 187) they say: “1,720 ancient and modern cattle from the 1000 Bull Genomes Project”. Which one is it? Are all these 1,720 individuals available on SRA/ENA? What about the genotypic/phenotypic data for all the individuals, maybe I missed something but I could only find some of the phenotypic data on the datagene.com.au website? The authors should upload the phenotypic and genotype data to a long term repository (e.g. dryad?) so that the exact same dataset can be accessed.

Author response: The whole-genome sequences of the 1000 Bull Genome Project (<http://www.1000bullgenomes.com/>) belong to global contributors and are available to participating members (including us). Access to the data requires consortium membership which is open. Depending on the individual members’ working progress and/or decision, some of the sequence data of the 1000 Bull Genome Project is publicly available and here is the latest information of publicly available 1000 Bull data (1832 samples): <https://www.ebi.ac.uk/eva/?eva-study=PRJEB42783>. We have also added Supplementary

Data 1 to label which 1720 samples are used. In the updated Supplementary Table 2, we also added the accession number of these samples that are publicly available.

The raw genotype and phenotype data of Australian dairy cattle are third-party data and are the private property of Australian farmers and Datagene. As we don't own the data, we don't have the right to publish the data. Access to these data requires permission from Datagene under a Data Use Agreement. This applies to any researchers who want to analyse their data, including ourselves.

Nonetheless, we aim to facilitate the validation of our results and promote data sharing. We uploaded 3 new summary datasets of those 7.9M sequence variants reported in the current manuscript. The first dataset is the summary of the effect direction and effect category of 7.9M variants where the ancestral alleles can be assigned (link:

<https://figshare.com/s/ef020d948523c31c0e67>). The 2nd dataset is the allele frequency of the mutant allele of those 7.9M sequence variants in Holstein and Jersey cattle from the 1000 Bull Genome Project (link: <https://figshare.com/s/20154b1d8e60e012e532>). These datasets should allow replicated results from the current manuscript and allow further use by other researchers. The 3rd data is the coordinates of conserved sites analysed in the current study.

To this end, the Data Availability statement has been updated: 'Our predictions of cattle ancestral alleles for those 42,573,455 sites have been made publicly available at:

<https://figshare.com/s/dd5985b76a413b56106b>. Multiple alignment data used to determine cattle ancestral alleles are publicly available via Ensembl EPO pipeline (http://asia.ensembl.org/info/genome/compara/multiple_genome_alignments.html).

Australian farmers and DataGene Australia (<http://www.datagene.com.au/>) are owners and custodians of the raw phenotype and genotype data of Australian dairy animals. Access to these data for research requires permission from DataGene under a Data Use Agreement. The DNA sequence data as part of the 1000 Bull Genomes Consortium [4, 6, 13] are available to consortium members and the membership is open. Sequence data of 1832 samples from the 1000 Bull Genome Project have been made publicly available at:

<https://www.ebi.ac.uk/eva/?eva-study=PRJEB42783>. The gene expression data is publically available (NCBI SRA SRP111067). In addition: 1. The summary data of the effect direction and effect category of those 7.9M sequence variants for which the ancestral alleles can be assigned is published at <https://figshare.com/s/ef020d948523c31c0e67>; 2. The allele frequency of mutant alleles of those 7.9M sequence variants for which the ancestral alleles can be assigned for the Holstein and Jersey cattle from the 1000 Bull Genome Project is published at <https://figshare.com/s/20154b1d8e60e012e532>; 3. The coordinates of conserved sites analysed in

the manuscript is published at: <https://figshare.com/s/df9d3662f8f7fb8e72da>. Other supporting data are shown in the supplementary materials of the current manuscript.’

Other comments:

The term mutant allele is not very common in the field of population genetics. Derived allele is a lot more common.

Author response: we think the two terms are equivalent and we have chosen the term of mutant allele. To address the reviewers’ comment, we noted this in the 2nd paragraph in Introduction: ‘Therefore the first objective of this study is to determine whether mutations, defined as the non-ancestral allele (**also known as derived alleles**) at segregating sites, tend to increase or decrease individual complex traits and whether this is depends on the trait’s association with fitness.’

Line 34: Cattle are artiodactyl.

Author response: this sentence has been revised as ‘... The cattle family diverged from **other** artiodactyls up to 30 million years ago...’

Line 55: The sentence that starts with “Genomics” does not follow the previous one.

Author response: this section has been revised as ‘A likely cause of directional dominance is that mutations tend to be deleterious and partially recessive. However, not all sites in the genome affecting a trait may show this pattern. Our second objective is to test the hypothesis that sites where the same allele has been conserved across vertebrate evolution are the most likely to show directional dominance. Therefore, we consider conserved sites and other polymorphic sites in this analysis.’

Line 60: “at 8M sequence variant sites on 37 traits of 4 breeds of over 113k modern cattle.”

This is very confusing, the variants on traits? Consider rephrasing.

Author response: this section has been re-written as: ‘In the present study, we use yak, sheep and camel as outgroup species to assign cattle ancestral alleles for 8M sequence variants (at 8M genomic sites). For each of these variants, the alternative to the ancestral allele is the mutant allele (MA). We estimate the effect of the mutant allele at these 8M variable sites on

37 traits of 113k cattle from 4 breeds.’

I am a bit concerned about the way that the author discretized the phastcons score. 0.9 seems a bit arbitrary? Is there any justification for this?

Author response: the following text has been added to the section ‘**Sequence variants under conserved sites across 100 vertebrate species.**’ of Methods: ‘Briefly, conservation was determined by the criteria of PhastCons score [18] > 0.9 based on the sequence data of those 100 species. The choice of 0.9 as the cutoff was arbitrary. However, since PhastCon score ranges from 0 to 1, this cutoff kept relatively highly conserved sites. Also, in a previous study [19], cattle variants from sites with PhastCon score > 0.9 were highly enriched for the heritability of cattle traits.’

The font of the figures are too small to read.

Author response: we have increased font size of all 4 main figures.

How did the author call aligned reads and called SNPs in the 98 (or 1.7k individuals) from the 1,000 bull genome project? How did they treat the ancient DNA data?

Author response: the 1000 Bull Genomes project has been running since 2012 and it has its standard pipeline to call variants (see details in [4, 7, 20] and <http://www.1000bullgenomes.com/>). As one of the consortium member, we receive all processed data and the data of those 98 or 1.7k individuals are just a subset of all the processed sequences.

The following text has been added to the 1st paragraph of the Methods (also added Supplementary Data 1): ‘The pre-processing of sequence reads and alignment of sequence data is done by partners using the standard 1000 Bull Genomes pipeline: <http://www.1000bullgenomes.com/>. Only BAM files from 1000 Bull Genomes partners are collected and processed by the consortium. The latest published data from the 1000 Bull Genomes (1832 samples) can be found at <https://www.ebi.ac.uk/eva/?eva-study=PRJEB42783>. The details of variant calling can be found in [7]. Briefly, Genome Analysis Toolkit (GATK v.3.8) [21] was used for variant calling. Variants from the GATK VQSR (Variant Quality Score Recalibration) 99.90 to 100.00 Tranche for SNP and INDEL were excluded, and Beagle v.4.0 [22] was used to impute sporadic missing.’

And the following has been added to the section of ‘**Mutant allele frequency and F_{ST} in different breeds/subspecies.**’ of Methods: ‘Additional information on these 1720 animals

including related accession numbers (if available) can be found in Supplementary Data 1. The ancient genome data were part of the project of Verdugo et al 2019 [23] who processed and published the original data (PRJEB31621 at European Nucleotide Archive). These data were collected by Run 7 of the 1000 Bull Genomes Project and processed by its standard pipeline (<http://www.1000bullgenomes.com/>).

Supplementary Table 2 should include individual accession numbers.

Author response: This table has now been updated with BiosampleID and BioprojectID from NCBI, if available.

References:

1. Jiang Y, Xie M, Chen W, Talbot R, Maddox JF, Faraut T, et al. The sheep genome illuminates biology of the rumen and lipid metabolism. *Science*. 2014;344(6188):1168-73. doi: 10.1126/science.1252806. PubMed PMID: 24904168; PubMed Central PMCID: PMC4157056.
2. Howie B, Fuchsberger C, Stephens M, Marchini J, Abecasis GR. Fast and accurate genotype imputation in genome-wide association studies through pre-phasing. *Nature genetics*. 2012;44(8):955.
3. Loh P-R, Danecek P, Palamara PF, Fuchsberger C, Reshef YA, Finucane HK, et al. Reference-based phasing using the Haplotype Reference Consortium panel. *Nature genetics*. 2016;48(11):1443.
4. Daetwyler HD, Capitan A, Pausch H, Stothard P, Van Binsbergen R, Brøndum RF, et al. Whole-genome sequencing of 234 bulls facilitates mapping of monogenic and complex traits in cattle. *Nature genetics*. 2014;46(8):858.
5. Rosen BD, Bickhart DM, Schnabel RD, Koren S, Elsik CG, Tseng E, et al. De novo assembly of the cattle reference genome with single-molecule sequencing. *GigaScience*. 2020;9(3):giaa021.
6. Hayes BJ, Daetwyler HD. 1000 Bull Genomes Project to Map Simple and Complex Genetic Traits in Cattle: Applications and Outcomes. *Annual review of animal biosciences*. 2019;7:89-102. Epub 2018/12/07. doi: 10.1146/annurev-animal-020518-115024. PubMed PMID: 30508490.
7. Daetwyler H, Brauning R, Chamberlain A, McWilliam S, McCulloch A, Vander Jagt C, et al., editors. 1000 Bull Genomes And Sheepgenomedb Projects: Enabling Costeffective Sequence Level Analyses Globally. *Proc Assoc Adv Anim Breed Genet*; 2017.
8. Pausch H, MacLeod IM, Fries R, Emmerling R, Bowman PJ, Daetwyler HD, et al. Evaluation of the accuracy of imputed sequence variant genotypes and their utility for causal variant detection in cattle. *Genetics Selection Evolution*. 2017;49(1):1-14.
9. Yao DW, O'Connor LJ, Price AL, Gusev A. Quantifying genetic effects on disease mediated by assayed gene expression levels. *Nature Genetics*. 2020;52(6):626-33.
10. Van Den Berg I, Hayes B, Chamberlain A, Goddard M. Overlap between eQTL and QTL associated with production traits and fertility in dairy cattle. *BMC genomics*. 2019;20(1):1-18.
11. Xiang R, Hayes BJ, Vander Jagt CJ, MacLeod IM, Khansefid M, Bowman PJ, et al. Genome variants associated with RNA splicing variations in bovine are extensively shared between tissues. *BMC Genomics*. 2018;19(1):521. doi: 10.1186/s12864-018-4902-8.
12. Yang J, Lee SH, Goddard ME, Visscher PM. GCTA: a tool for genome-wide complex trait analysis. *The American Journal of Human Genetics*. 2011;88(1):76-82.
13. Bouwman AC, Daetwyler HD, Chamberlain AJ, Ponce CH, Sargolzaei M, Schenkel FS, et al. Meta-analysis of genome-wide association studies for cattle stature identifies common genes that regulate body size in mammals. *Nature genetics*. 2018;50(3):362.

14. Meuwissen T, Hayes B, Goddard M. Prediction of total genetic value using genome-wide dense marker maps. *Genetics*. 2001;157(4):1819-29.
15. Mehlhoff JD, Stearns FW, Rohm D, Wang B, Tsou E-Y, Dutta N, et al. Collateral fitness effects of mutations. *Proceedings of the National Academy of Sciences*. 2020;117(21):11597-607.
16. Keller LF, Waller DM. Inbreeding effects in wild populations. *Trends in Ecology & Evolution*. 2002;17(5):230-41. doi: [https://doi.org/10.1016/S0169-5347\(02\)02489-8](https://doi.org/10.1016/S0169-5347(02)02489-8).
17. Turelli M, Ginzburg LR. Should individual fitness increase with heterozygosity? *Genetics*. 1983;104(1):191-209. PubMed PMID: 6862183.
18. Siepel A, Bejerano G, Pedersen JS, Hinrichs AS, Hou M, Rosenbloom K, et al. Evolutionarily conserved elements in vertebrate, insect, worm, and yeast genomes. *Genome research*. 2005;15(8):1034-50.
19. Xiang R, Berg Ivd, MacLeod IM, Hayes BJ, Prowse-Wilkins CP, Wang M, et al. Quantifying the contribution of sequence variants with regulatory and evolutionary significance to 34 bovine complex traits. *Proceedings of the National Academy of Sciences*. 2019;116(39):19398-408. doi: 10.1073/pnas.1904159116.
20. Daetwyler H, Xiang R, Yuan Z, Bolormaa S, Vander Jagt C, Hayes B, et al. Integration of functional genomics and phenomics into genomic prediction raises its accuracy in sheep and dairy cattle. *Proceedings of the Association for the Advancement of Animal Breeding and Genetics*, Armidale, NSW, Australia. 2019:11-4.
21. DePristo MA, Banks E, Poplin R, Garimella KV, Maguire JR, Hartl C, et al. A framework for variation discovery and genotyping using next-generation DNA sequencing data. *Nature genetics*. 2011;43(5):491-8.
22. Browning SR, Browning BL. Rapid and accurate haplotype phasing and missing-data inference for whole-genome association studies by use of localized haplotype clustering. *American journal of human genetics*. 2007;81(5):1084-97. Epub 2007/10/10. doi: 10.1086/521987. PubMed PMID: 17924348; PubMed Central PMCID: PMC2265661.
23. Verdugo MP, Mullin VE, Scheu A, Mattiangeli V, Daly KG, Delser PM, et al. Ancient cattle genomics, origins, and rapid turnover in the Fertile Crescent. *Science*. 2019;365(6449):173-6.

Reviewers' comments:

Reviewer #1 (Remarks to the Author):

The authors have adequately addressed my concerns as well as the comments from the other two referees. The current version is acceptable for publication after language and some format correction.

Reviewer #2 (Remarks to the Author):

The revised version of the manuscript is a huge improvement on the previous submission. I believe the research activities on which the manuscript is based are of a very high standard, and represent a substantial body of work. However, the manuscript is only of an average standard. It reports numerous analyses, that are presumably very familiar to the authors, but the subtleties of the similarities and differences of those analyses in terms of datasets, traits, sites, and alleles are difficult for the reader to follow unambiguously without multiple end-to-end reads. I believe this manuscript could have a significant number of citations and achieve a high impact, but will not reach its potential in its current form.

Title – I wonder if “Mutant alleles differentially shape fitness and other complex traits” would be more appropriate. The current version suggests fitness is not a complex trait.

Abstract – This should stand alone, and will be the only thing that many readers look at. I don't believe it does a good job of summarising the manuscript. In the first sentence on line 11, I am not sure what “Classical mutant alleles (MAs) with large effects on phenotype....” actually are. Do the authors mean “Mutant alleles that have been classically recognised have large effects on ...”? The first sentence leads me to believe the definition of MA in the paper is a “classical....large effect...allele”. Later it is apparent that an MA does not have to have any effect. Line 14 presumably “100 vertebrate species” rather than 100 vertebrates. Line 14 reports the first major finding of the paper – that “heterozygosity ...at conserved sites...increase...”. Taken as a whole, that finding is only a minor part of the manuscript in its entirety. But it's a major component of the abstract. The sentence on line 19-21 is awkward, and does not seem to be grammatically correct. The last sentence is also awkward “bias in mutation towards alleles that are selected against.” I think the abstract could easily be greatly improved. After the title, it is the first signpost detailing what the reader should be expecting in the rest of the manuscript. I don't believe it does this very well in its current form.

There are many grammatical errors, spelling mistakes and awkward lines throughout the manuscript. The bracketed item in lines 27-28 is but one example. Line 30 “showd”. Line 31 effects.

Line 37-39 is speculation. The claimed advantage has not yet been demonstrated. This manuscript does not demonstrate it. In practice genomic selection as used in a variety of industries and applications use only anonymous markers, rather than mutations that are causal in any respect.

Line 49 What are “genomic sites on traits of cattle”. I thought genomic sites refers to the genome. Do the authors mean “we introduce a test for directional dominance on traits of cattle by estimating the heterozygosity at genomic sites”?

Line 71. “We estimate the effect of the mutant allele” is singular. What effect of the mutant allele? The additive effect? The dominance effect? Later it seems they consider both. DO the authors mean “We estimate the effects of the genotypes carrying the mutant allele”?

Line 73 the “collective” effect rather than effect. Also “both...and” should be “either...or”

Line 76 missing space between words

Line 90 A “subset” – do you mean “A subset of about half of these ”

Line 93 vertebrate "species"

Line 92 Do you mean "we fit a covariate representing the average heterozygosity"

Line 95 Spelling

Line 98 and 99. Presumably there were two covariates fitted, one representing heterozygosity at conserved sites and the other representing the heterozygosity at sites that were not conserved. If I have misunderstood, it speaks to the materials and methods being inadequate. If I have understood correctly, I would be interested to have known the covariance between the heterozygosity that was classified in this way. According to the abstract, the conclusion relating to heterozygosity is one of the main findings of the paper, but the details leave me unfulfilled.

Line 102 "exclusively" seems a bit strong. It means that none of the loci in the non conserved regions explain inbreeding depression or heterosis. The manuscript does not show enough of the analysis of that to convince me of the authors conclusion.

Figure 1 "rest sites" is not clear English.

Line 108 "fitting a covariate representing H'" and line 109 "another covariate representing H'" from the remaining "

Line 133-134 the effect of an allele is opposite in sign to the effect of the other allele. This implies a certain kind of fitting of the "effect of the allele" which was not described earlier.

Line 141 "Here the effect size ...which is inversely related to the p-value. That is true, but I am more interested in the effect size than the p-value in analyses with such large numbers of individuals. I would have liked to have been told something about what "large" and "small" means in an informative manner – such as in relation to phenotypic or genetic standard deviations which would still be able to be interpreted without familiarity of the particular trait units etc.

Line 144 "independent" is not the right word. The same phenotypes may have contributed to the yield deviation on cows and the daughter yield deviations on their sires. Analysing the sexes separately is not an independent analysis. Unless there were additional edits of the animals and/or data to ensure this was the case. The materials and methods did not provide additional information in that regard.

Line 258 "remembering" is not the correct word. Many readers would never have heard that these traits are negatively correlated so could hardly remember it.

Line 377 don't need "new" twice

Line 385 "and crops" seems like a stretch.

Line 394 Poor clarity

Line 395 "of 98 breeds, one individual per breed"

Line 396-7 Delete

Line 407 -409 Poor clarity. Missing what?

Line 415 the software wasn't "from" python

Line 428 – the software does not "make requests"

Line 431 "up to 1". I don't follow.

Line 447 Poor clarity. Sites are not "highly enriched for the heritability of cattle traits".

Line 455 daughter trait deviations is very vague. Were they really just the average of the trait deviations of the daughters? Surely not. The data is from an admixed population. Is the daughter trait deviation not adjusted for the merit (and breed) of the mate?

Line 464 interpretability is awkward.

Line 494 independently.

Line 507 is a little misleading. While the test of the effect in the GWAS may have been one at a time, the methods claimed that in the GWAS analysis a G matrix was fitted simultaneously that "was based on all variants".

Line 527-30. But the effects of the alternate alleles are simply opposite in sign, as the authors stated earlier. This wording on comparing the effect distribution needs clarification in that regard.

Line 670 publically!

Line 681 awkward sentence

Overall, the paper represents a substantial effort, and comprises very good science. The findings are novel, and will be of interest to others in the community, and the wider field. The statistical analyses are appropriate and valid, other than the minor comments I have made above. The paper may influence thinking in the field, but is more likely to do this if the abstract was rewritten and the paper was further edited to improve its readability by those who don't want to have to read the manuscript multiple times to try and untangle the specifics of the datasets sites, traits, etc

Response to reviewers' comments

Revised COMMSBIO-21-1004-A: 'Mutant alleles differentially shape cattle complex traits and fitness'

We thank the reviewers for their time and effort in providing comments and suggestions on our manuscript. Please find our point-to-point response to reviewers' comments in blue text in the following.

Reviewers' comments:

Reviewer #1 (Remarks to the Author):

The authors have adequately addressed my concerns as well as the comments from the other two referees. The current version is acceptable for publication after language and some format correction.

Author response: We thank the reviewer for positive comments. We have improved the language and the manuscript in light of comments from all rereviews.

Reviewer #2 (Remarks to the Author):

The revised version of the manuscript is a huge improvement on the previous submission. I believe the research activities on which the manuscript is based are of a very high standard, and represent a substantial body of work. However, the manuscript is only of an average standard. It reports numerous analyses, that are presumably very familiar to the authors, but the subtleties of the similarities and differences of those analyses in terms of datasets, traits, sites, and alleles are difficult for the reader to follow unambiguously without multiple end-to-end reads. I believe this manuscript could have a significant number of citations and achieve a high impact, but will not reach its potential in its current form.

Author response: we thank the reviewer for providing us with detailed feedback. Please see our responses/revisions according to your comments in the following text.

Title – I wonder if “Mutant alleles differentially shape fitness and other complex traits” would be more appropriate. The current version suggests fitness is not a complex trait.

Author response: the title has been revised to what the reviewer suggested.

Abstract – This should stand alone, and will be the only thing that many readers look at. I don't believe it does a good job of summarising the manuscript. In the first sentence on line 11, I am not sure what “Classical mutant alleles (MAs) with large effects on phenotype....” actually are. Do the authors mean “Mutant alleles that have been classically recognised have large effects on ...”? The first sentence leads me to believe the definition of MA in the paper

is a “classical....large effect...allele”. Later it is apparent that an MA does not have to have any effect. Line 14 presumably “100 vertebrate species” rather than 100 vertebrates. Line 14 reports the first major finding of the paper – that “heterozygosity ...at conserved sites...increase...”. Taken as a whole, that finding is only a minor part of the manuscript in its entirety. But it’s a major component of the abstract. The sentence on line 19-21 is awkward, and does not seem to be grammatically correct. The last sentence is also awkward “bias in mutation towards alleles that are selected against.” I think the abstract could easily be greatly improved. After the title, it is the first signpost detailing what the reader should be expecting in the rest of the manuscript. I don’t believe it does this very well in its current form.

Author response: we thank the reviewer’s suggestion on the first sentence and we have revised the sentence according to the suggestion.

‘vertebrates’ is changed to ‘vertebrate species’.

The sentence at lines 19-21 is revised as: ‘However, the frequency of MAs decreasing stature and fat and protein concentration, increasing gestation length and somatic cell count were lower than the frequency of MAs with the opposite effect.’

The last sentence is revised as: ‘Taken together, our results imply two classes of genomic sites subject to long-term selection: sites conserved across vertebrates show hybrid vigour while sites subject to less long-term selection show a bias in mutation towards undesirable alleles.’

There are many grammatical errors, spelling mistakes and awkward lines throughout the manuscript. The bracketed item in lines 27-28 is but one example. Line 30 “showd”. Line 31 effects.

Author response: we have carefully revised the text of the manuscript. The 1st sentence of the introduction has been revised and bracketed item in lines 27-28 is removed.

‘showd’ is changed to ‘showed’.

‘effect’ is changed to ‘effects’.

Line 37-39 is speculation. The claimed advantage has not yet been demonstrated. This manuscript does not demonstrate it. In practice genomic selection as used in a variety of industries and applications use only anonymous markers, rather than mutations that are causal in any respect.

Author response: we apologize for missing a reference for using causal variants in genomic prediction. This sentence is now revised as: ‘Genomic selection, which is widely used in animal breeding ¹, has been demonstrated to be enhanced by fitting variants with biological priors ². Therefore, it may be an advantage to also know a priori whether mutations are more likely to increase or decrease traits of interest.’

Line 49 What are “genomic sites on traits of cattle”. I thought genomic sites refers to the genome. Do the authors mean “we introduce a test for directional dominance on traits of cattle by estimating the heterozygosity at genomic sites?”

Author response: we thank the suggestion from the reviewer and we have changed the sentence according to the suggestion.

Line 71. “We estimate the effect of the mutant allele” is singular. What effect of the mutant allele? The additive effect? The dominance effect? Later it seems they consider both. DO the authors mean “We estimate the effects of the genotypes carrying the mutant allele”?

Author response: we thank the suggestion from the reviewer. This part of the sentence has been revised to ‘We estimate the effects of the genotypes based on the mutant allele’.

Line 73 the “collective” effect rather than effect. Also “both...and” should be “either...or”

Author response: this sentence has been revised to what the reviewer suggested.

Line 76 missing space between words

Author response: a space is added between ‘frequency’ and ‘compared’.

Line 90 A “subset” – do you mean “A subset of about half of these “

Author response: this sentence has been revised to what the reviewer suggested.

Line 93 vertebrate “species”

Author response: ‘vertebrates’ is changed to ‘vertebrate species.’

Line 92 Do you mean “we fit a covariate representing the average heterozygosity”

Author response: this part of the sentence is changed to ‘we fit covariates of ...’

Line 95 Spelling

Author response: this error is fixed.

Line 98 and 99. Presumably there were two covariates fitted, one representing heterozygosity at conserved sites and the other representing the heterozygosity at sites that were not conserved. If I have misunderstood, it speaks to the materials and methods being inadequate. If I have understood correctly, I would be interested to have known the covariance between the heterozygosity that was classified in this way. According to the abstract, the conclusion relating to heterozygosity is one of the main findings of the paper, but the details leave me unfulfilled.

Author response: the reviewer understood the method correctly. We agree with the reviewer that additional work of testing the interaction between the two covariates is interesting. However, the existing analysis of heterozygosity supports our conclusions that 1) heterozygosity from conserved sites, instead of non-conserved sites, significantly contributes to traits and 2) such contribution allows us to identify traits that are related to fitness. Also, as

the reviewer pointed out in the comments, the manuscript already ‘represents a substantial effort’, ‘reports numerous analyses’ and ‘the statistical analyses are appropriate and valid’. Therefore, we wish to pursue the additional idea in future work.

Line 102 “exclusively” seems a bit strong. It means that none of the loci in the non conserved regions explain inbreeding depression or heterosis. The manuscript does not show enough of the analysis of that to convince me of the authors conclusion.

Author response: we have replaced the word ‘exclusively’ with ‘predominantly’ in this sentence.

Figure 1 “rest sites” is not clear English.

Author response: ‘rest sites’ is changed to ‘non-conserved sites’ in Figure 1.

Line 108 “fitting a covariate representing H’ “ and line 109 “another covariate representing H’ from the remaining “

Author response: we have revised this sentence according to the suggestion of the reviewer.

Line 133-134 the effect of an allele is opposite in sign to the effect of the other allele. This implies a certain kind of fitting of the “effect of the allele” which was not described earlier.

Author response: since this sentence is more related to methods and may be confusing to be in results, we have removed this sentence to the method section.

Line 141 “Here the effect size ...which is inversely related to the p-value. That is true, but I am more interested in the effect size than the p-value in analyses with such large numbers of individuals. I would have liked to have been told something about what “large” and “small” means in an informative manner – such as in relation to phenotypic or genetic standard deviations which would still be able to be interpreted without familiarity of the particular trait units etc.

Author response: To address the comments from the reviewer we conducted additional analysis to quantify trait variance explained by variants with small-, medium- and large effects. The following sentence is added to the 2nd paragraph of the section ‘Biases in trait effects between ancestral and mutant alleles’: On average, each variant from the large, medium and small-effect category explained 0.31% ($\pm 0.043\%$), 0.07% ($\pm 0.009\%$) and 0.015% ($\pm 0.0004\%$) of the variance in cow traits, respectively (Supplementary Table 3).

Line 144 “independent” is not the right word. The same phenotypes may have contributed to the yield deviation on cows and the daughter yield deviations on their sires. Analysing the sexes separately is not an independent analysis. Unless there were additional edits of the animals and/or data to ensure this was the case. The materials and methods did not provide additional information in that regard.

Author response: we replaced the word ‘independent’ with ‘different’.

Line 258 “remembering” is not the correct word. Many readers would never have heard that these traits are negatively correlated so could hardly remember it.

Author response: This sentence has been revised to ‘Note that milk yield is negatively correlated with fat% and protein% ($r = -0.83$ and -0.78 , respectively).’

Line 377 don’t need “new” twice

Author response: the ‘new’ next to hypothesis is removed.

Line 385 “and crops” seems like a stretch.

Author response: ‘crops’ is replaced by ‘other species.’

Line 394 Poor clarity

Author response: this sentence has been changed to ‘Camel without a rumen is distantly related to ruminant cattle, as they are artiodactyls.’

Line 395 “of 98 breeds, one individual per breed”

Line 396-7 Delete

Author response: These two sentences are revised according to the suggestion from the reviewer.

Line 407 -409 Poor clarity. Missing what?

Author response: this sentence is revised as ‘Variants from the GATK VQSR (Variant Quality Score Recalibration) 99.90 to 100.00 Tranche for SNP and INDEL were excluded, and Beagle v.4.0 36 was used to impute variants with sporadic missing.’

Line 415 the software wasn’t “from” python

Author response: ‘from’ is replaced by ‘based on’.

Line 428 – the software does not “make requests”

Author response: ‘requested’ is replaced by ‘required’.

Line 431 “up to 1”. I don’t follow.

Author response: This sentences has been revised as ‘For each outgroup species, the total allele count for genome sequences at each site was up to 1.’. Please note that this is a software setting as instructed by its user manual (<https://sourceforge.net/projects/est-usfs/>). Please refer questions regarding the software to its writer: Keightley et al³.

Line 447 Poor clarity. Sites are not “highly enriched for the heritability of cattle traits”.

Author response: To improve the clarity we have added '(occupying 2% of the genome to explain up to 42% heritability of traits)' at the end of this sentence.

Line 455 daughter trait deviations is very vague. Were they really just the average of the trait deviations of the daughters? Surely not. The data is from an admixed population. Is the daughter trait deviation not adjusted for the merit (and breed) of the mate?

Author response: we apologize for the ambiguity of this part. This section has been updated in the revised manuscript.

Line 464 interpretability is awkward.

Author response: we have broken this long sentence into 2 sentences.

Line 494 independently.

Line 507 is a little misleading. While the test of the effect in the GWAS may have been one at a time, the methods claimed that in the GWAS analysis a G matrix was fitted simultaneously that "was based on all variants".

Author response: we have removed the word 'independently'. The GRM is built using all variants, but in the end it is the animal relationships that are fitted as the random effects with other fixed effects including the SNP effect in the linear model.

Line 527-30. But the effects of the alternate alleles are simply opposite in sign, as the authors stated earlier. This wording on comparing the effect distribution needs clarification in that regard.

Author response: to improve the clarity we have added: 'For example, for a given set of variants, if their mutant alleles had a bias in effect direction towards increasing the trait, their ancestral alleles would have a bias in effect direction towards decreasing the trait. This then would create a difference in effect distribution between mutant and ancestral alleles.'

Line 670 publically!

Author response: this is fixed.

Line 681 awkward sentence

Author response: this sentence has been revised as 'The probability of ancestral allele assignment was estimated using the software est-sfs published by Keightley et al 2018³.'

Overall, the paper represents a substantial effort, and comprises very good science. The findings are novel, and will be of interest to others in the community, and the wider field. The statistical analyses are appropriate and valid, other than the minor comments I have made above. The paper may influence thinking in the field, but is more likely to do this if the abstract was rewritten and the paper was further edited to improve its readability by those

who don't want to have to read the manuscript multiple times to try and untangle the specifics of the datasets sites, traits, etc

Author response: We thank the reviewer for detailed comments on the manuscript. We do appreciate the advice to improve the clarity of the manuscript and have done the revisions accordingly.

- 1 Meuwissen, T., Hayes, B. & Goddard, M. Genomic selection: A paradigm shift in animal breeding. *Animal frontiers* **6**, 6-14 (2016).
- 2 Xiang, R. *et al.* Genome-wide fine-mapping identifies pleiotropic and functional variants that predict many traits across global cattle populations. *Nature Communications* **12**, 860, doi:10.1038/s41467-021-21001-0 (2021).
- 3 Keightley, P. D. & Jackson, B. C. Inferring the probability of the derived vs. the ancestral allelic state at a polymorphic site. *Genetics* **209**, 897-906 (2018).